# Environmental DNA from archived leaves reveals widespread temporal turnover and biotic homogenization in forest arthropod communities

Henrik Krehenwinkel[1][*][†], Sven Weber[1][†], Rieke Broekmann[1], Anja Melcher[1], Julian Hans[1], Rüdiger Wolf[1], Axel Hochkirch[1], Susan Rachel Kennedy[1], Jan Koschorreck[2], Sven Künzel[3], Christoph Müller[4], Rebecca Retzlaff[1], Diana Teubner[1], Sonja Schanzer[4], Roland Klein[1], Martin Paulus[1], Thomas Udelhoven[1], Michael Veith[1]

[1]University of Trier, Trier, Germany; [2]German Federal Environment Agency, Berlin, Germany; [3]Max Planck Institute for Evolutionary Biology, Plön, Germany; [4]Ludwig Maximilians University, Munich, Germany

*For correspondence: krehenwinkel@uni-trier.de

†These authors contributed equally to this work

**Abstract** A major limitation of current reports on insect declines is the lack of standardized, long-term, and taxonomically broad time series. Here, we demonstrate the utility of environmental DNA from archived leaf material to characterize plant-associated arthropod communities. We base our work on several multi-decadal leaf time series from tree canopies in four land use types, which were sampled as part of a long-term environmental monitoring program across Germany. Using these highly standardized and well-preserved samples, we analyze temporal changes in communities of several thousand arthropod species belonging to 23 orders using metabarcoding and quantitative PCR. Our data do not support widespread declines of $\alpha$-diversity or genetic variation within sites. Instead, we find a gradual community turnover, which results in temporal and spatial biotic homogenization, across all land use types and all arthropod orders. Our results suggest that insect decline is more complex than mere $\alpha$-diversity loss, but can be driven by $\beta$-diversity decay across space and time.

## Editor's evaluation

This landmark study reveals novel temporal arthropod biodiversity insights that can be leveraged from environmental DNA traces, that have been cryopreserved on leaf tissue as part of a long-term monitoring scheme. The strength of the evidence underlying the major conclusions is convincing and limitations in the quantitative aspects of the data synthesis are acknowledged appropriately. The work will be of interest to a breadth of ecological practitioners.

## Introduction

Dramatic declines of terrestrial insects have been reported in recent years, particularly in areas of intensified land use (*Hallmann et al., 2017*; *van Klink et al., 2020*; *Seibold et al., 2019*; *Sánchez-Bayo and Wyckhuys, 2019*). However, some authors have urged caution in generalizing these results (*Didham et al., 2020*; *Thomas et al., 2019*; *Cardoso et al., 2019*), suggesting that reported patterns of decline may be more localized than currently assumed or reflect long-term natural abundance fluctuations (*Macgregor et al., 2019*; *Crossley et al., 2020*). Most studies on insect decline suffer from a

**eLife digest** Insects are a barometer of environmental health. Ecosystems around the world are being subjected to unprecedented man-made stresses, ranging from climate change to pollution and intensive land use. These stresses have been associated with several recent, dramatic declines in insect populations, particularly in areas with heavily industrialised farming practices.

Despite this, the links between insect decline, environmental stress, and ecosystem health are still poorly-understood. A decline in one area might look catastrophic, but could simply be part of normal, longer-term variations. Often, we do not know whether insect decline is a local phenomenon or reflects wider environmental trends. Additionally, most studies do not go far back enough in time or cover a wide enough geographical range to make these distinctions.

To understand and combat insect decline, we therefore need reliable methods to monitor insect populations over long periods of time. To solve this problem, Krehenwinkel, Weber et al. gathered data on insect communities from a new source: tree leaves. Originally, these samples were collected to study air pollution, but they also happen to contain the DNA of insects that interacted with them before they were collected – for example, DNA deposited in chew marks where the insects had nibbled on the leaves. This is called environmental DNA, or eDNA for short.

To survey the insect communities that lived in these trees, Krehenwinkel, Weber et al. first extracted eDNA from the leaves and sequenced it. Analysis of the different DNA sequences from the leaf samples revealed not only the number of insect species, but also the abundance (or rarity) of each species within each community. Importantly, the leaves had been collected and stored in stable conditions over several decades, allowing changes in these insect populations to be tracked over time.

eDNA analysis revealed subtle changes in the make-up of forest insect communities. In the forests where the leaves were collected, the total number of insect species remained much the same over time. However, many individual species still declined, only to be replaced by newcomer species. These 'colonisers' are also widespread, which will likely lead to an overall pattern of fewer species that are more widely distributed – in other words, more homogeneity.

The approach of Krehenwinkel, Weber et al. provides a reliable method to study insect populations in detail, over multiple decades, using archived samples from environmental studies. The information gained from this has real-world significance for environmental issues with enormous social impact, ranging from conservation, to agriculture and even public health.

lack of long-term time series data and are limited in geographic and taxonomic breadth, often using biomass as a proxy for diversity estimates (*Hallmann et al., 2017*; *Daskalova et al., 2021*). Hence, what is needed are methods and sample types that yield standardized long-term time series data for the diversity of arthropod communities across broad taxonomic and geographic scales (*Forister et al., 2021*).

In recent years, environmental DNA metabarcoding has offered a promising new approach to monitor biological communities (*Taberlet et al., 2018*; *Tautz et al., 2002*; *Thomsen and Sigsgaard, 2019*). This includes terrestrial arthropods, whose eDNA can be recovered from various substrates, for example plant material (*Nakamura et al., 2017*). Here, we develop a DNA metabarcoding and quantitative PCR (qPCR) protocol to simultaneously recover diversity and relative DNA copy number of arthropod community DNA from powdered leaf samples. We then analyze 30-year time series data of arthropod communities from canopy leaf material of four tree species from 24 sites across Germany. These sites represent four land use types with different degrees of anthropogenic disturbance: urban parks, agricultural areas, timber forests, and national parks (*Figure 1*). The samples were collected using a highly standardized protocol by the German Environmental Specimen Bank (ESB), a large biomonitoring effort for Germany's ecosystems, and stored at below −150°C. By basing our analyses on DNA sequences, we can measure diversity from haplotype variation within species to taxonomic diversity of the whole community.

Current studies on insect decline primarily focus on site-based assessments of $\alpha$-diversity and biomass (*Hallmann et al., 2017*; *Seibold et al., 2019*). Yet, these metrics alone are insufficient for characterizing ongoing biodiversity change (*Marta et al., 2021*; *Kortz and Magurran, 2019*; *Magurran and Henderson, 2010*). Significant temporal community change and declines can also

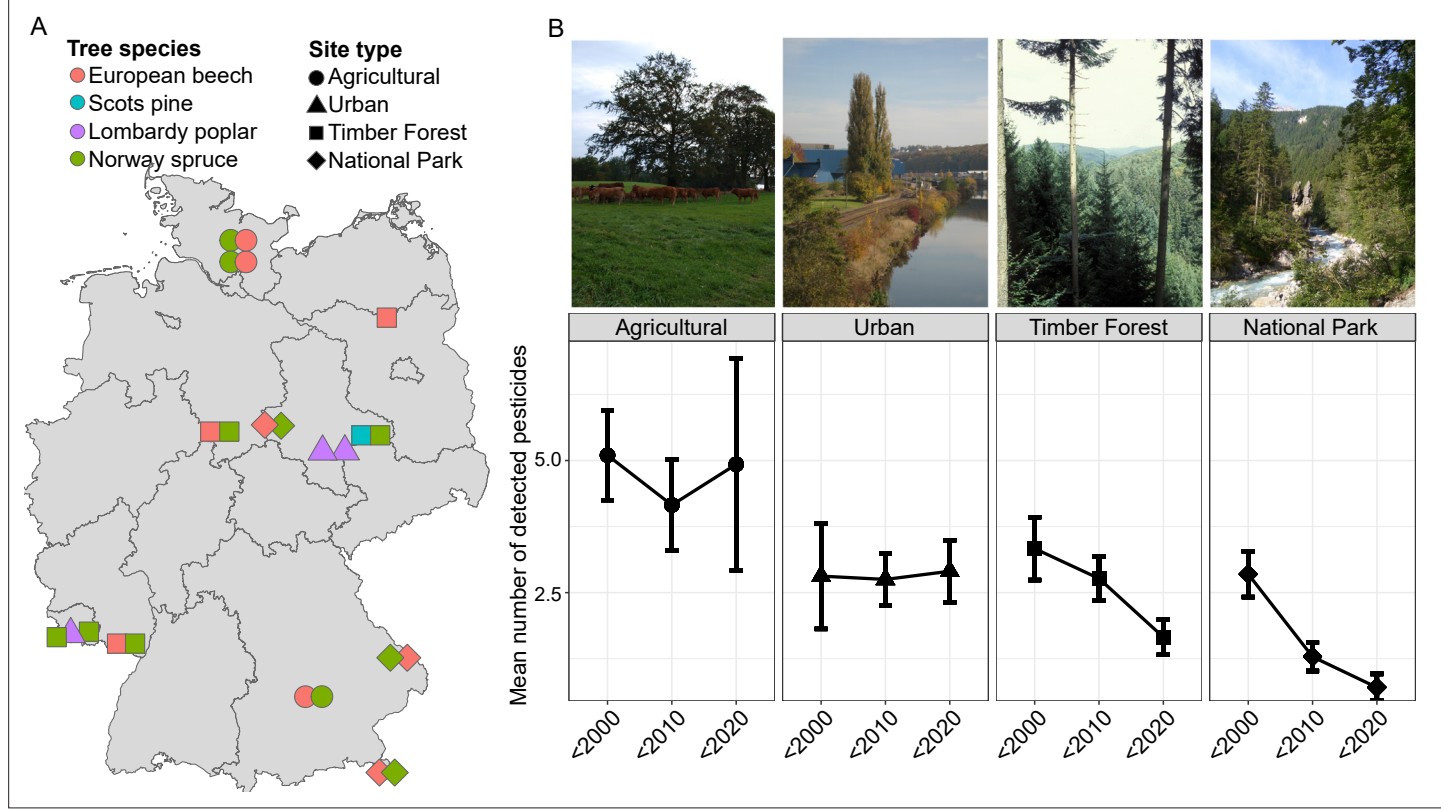

**Figure 1.** Overview of different sampling sites and samples in this study. (**A**) Sampling sites, land use types, and the four tree species (European Beech *Fagus sylvatica* (N = 98), Lombardy Poplar *Populus nigra* 'italica' (N = 65), Norway Spruce *Picea abies* (N = 123), and Scots Pine *Pinus sylvestris* (N = 26)) sampled by the ESB in Germany. (**B**) Representative images of sampling sites for the four land use types and average number of detected pesticides (and 95 % confidence interval) across these land use types in three time periods (before 2000, 2000–2009, and 2010–2018). Gas chromatography–tandem mass spectrometry (GC–MS/MS) analysis (*Löbbert et al., 2021*) shows that the detected pesticide load distinguishes the different land use types, with agricultural sites continuously showing the highest number of pesticides.

occur at the scale of $\beta$- or $\gamma$-diversity, without affecting local richness. This may be driven by community turnover and spatial biotic homogenization (*Karp et al., 2012*). But diversity may also vary temporally. Fluctuating occurrences of transient species considerably increase diversity within single sites over time (*D'Souza and Hebert, 2018*). The loss of such transient species in favor of taxa with a temporally stable occurrence results in an increasingly predictable community and hence a temporal diversity decline. Biotic turnover may occur gradually following changing environmental conditions, but can also occur abruptly, when ecosystems reach tipping points (*Barnosky et al., 2012*). In the latter case, a rapid and considerable biotic remodeling of the ecosystem may be found. The relevance of these spatial and temporal factors in insect decline remains largely elusive.

Here, we use our high-resolution data on arthropods from canopy leaf samples to test the hypotheses that (1) $\alpha$-diversity and biomass of canopy-associated arthropod communities have declined in the last 30 years (*Hallmann et al., 2017*; *Seibold et al., 2019*), or (2) community change has occurred in the form of turnover and possibly homogenization of communities across space and time (*Kortz and Magurran, 2019*; *Dornelas et al., 2014*; *Thomsen et al., 2016*). Last, we hypothesize that (3) biodiversity declines will be particularly pronounced in areas of intensified land use (*Outhwaite et al., 2022*).

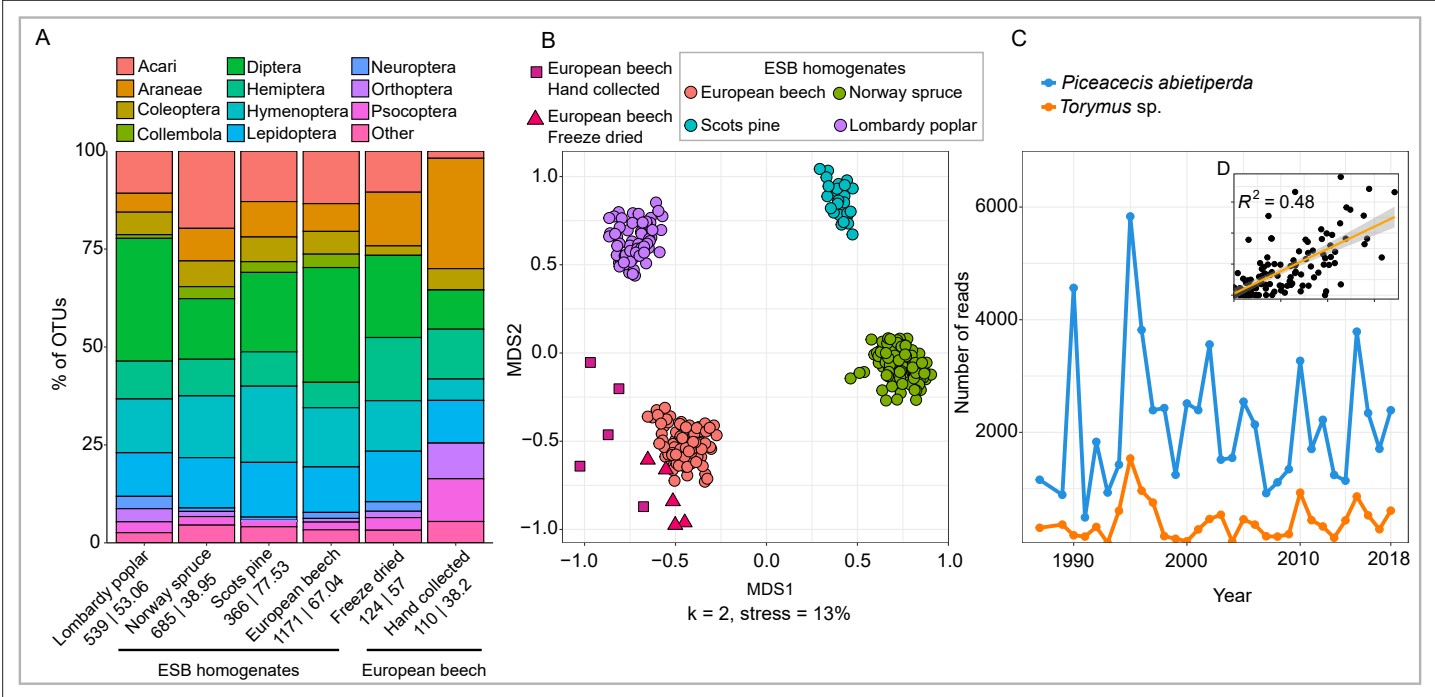

**Figure 2.** Recovery of diversity and interactions in canopy-associated arthropod communities from leaf material. (**A**) Barplot showing recovered order composition of OTUs across the four tree species (N = 312). In addition to ESB samples, results from freeze-dried leaves stored at room temperature for 6–8 years (N = 5) and hand-collected bulk insect samples (N = 5) from European Beech are shown. Orders amounting to less than 1% of the total OTU number are merged as 'Other'. Numbers below each barplot show the total number of arthropod OTUs followed by the mean OTU number per sample. (**B**) Non-metric multidimensional scaling (NMDS) plot showing tree-specific composition of arthropod communities for the same samples. (**C**) Temporal changes in abundance of the spruce gall midge *Piceacecis abietiperda* and its parasitoid *Torymus* sp. between 1987 and 2018 in a spruce forest in the Saarland. Inset (D) shows correlation of relative abundance between the two species across all ESB spruce samples.

The online version of this article includes the following figure supplement(s) for figure 2:

**Figure supplement 1.** Effect of replication and sample input on recovered arthropod diversity.

**Figure supplement 2.** Effect of weight of plant material and precipitation events before sampling on the recovered arthropod diversity.

**Figure supplement 3.** Contamination check in the cryomill.

**Figure supplement 4.** Rarefaction curves for all analyzed samples.

**Figure supplement 5.** Comparison of recovered taxonomic composition and diversity patterns for the two COI markers (ZBJ-ArtF1c/ZBJ-ArtR2c vs. NoPlantF_270/mICOIintR_W) used in this study.

**Figure supplement 6.** Ecological diversity of arthropod species recovered from the four tree species.

**Figure supplement 7.** NMDS showing arthropod community differentiation by site, separated by tree species.

## Results

### A standardized protocol to characterize plant-associated arthropod communities

We developed a standardized and robust protocol to reproducibly recover plant-associated arthropod communities from powdered leaf material. We controlled for effects of the amount of leaf material per sample, rainfall before sampling, amount of leaf homogenate used for DNA extraction, extraction replication, and primer choice on the recovered diversity (see Methods and *Figure 2—figure supplements 1–5* for details on standardization).

Using our optimized protocol, we analyzed 312 ESB leaf samples. We recovered 2054 OTUs from our samples, with different tree species having significantly different OTU numbers on average (*Figure 2A*, linear mixed model [LMM], p < 0.05). Nevertheless, all tree species showed a balanced and relatively similar taxonomic composition at the order level (*Figure 2A*, *Figure 2—figure supplement 5E*). We identified 23 orders, 218 families, and 413 genera. The richest order was Diptera (600 OTUs in 48 families), followed by Hymenoptera (369 OTUs in 21 families), Acari (293 OTUs in 21

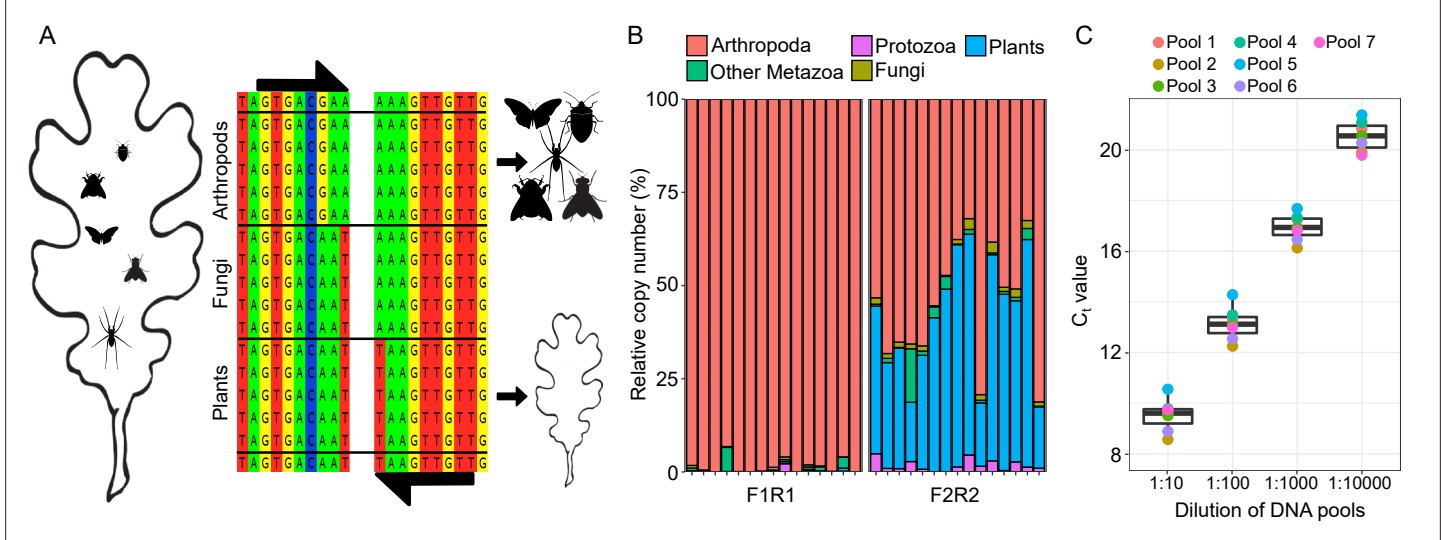

**Figure 3.** Quantitative PCR (qPCR) experiment to detect relative rDNA copy number of arthropods in plant homogenates. (**A**) Schematic overview of the blocking approach to amplify homologous 18SrDNA fragments for either arthropod or plant DNA, based on lineage-specific priming mismatches. (**B**) Effect of primer mismatches on the recovery of arthropod sequences. Barplots show recovered read proportion of different higher taxa from 15 ESB leaf samples. The left plot shows the effect of a diagnostic mismatch in forward and reverse primers, while the right plot shows the effect of only a forward primer mismatch. (**C**) Boxplot showing CT values recovered from the seven mock communities of arthropod species from 13 different orders (see *Figure 3—figure supplement 1* for community composition) and across a 1.10000 dilution series. Separate CT values for each community are indicated by the dots (*Figure 3—figure supplement 1*).

The online version of this article includes the following figure supplement(s) for figure 3:

**Figure supplement 1.** Taxonomic composition of the mock communities used to test the qPCR assay.

families), Lepidoptera (233 OTUs in 32 families), Hemiptera (152 OTUs in 19 families), Coleoptera (133 OTUs in 29 families), and Araneae (99 OTUs in 15 families). The recovered species assemblages were ecologically diverse, including herbivores, detritivores, predators, parasites, and parasitoids (*Figure 2—figure supplement 6*). Each tree species harbored a unique arthropod community (*Figure 2B*, *Figure 2—figure supplement 5C, D*), with typical monophagous taxa exclusively recovered from their respective host trees. The arthropod communities from different sites and land use types were also differentiated within tree species (*Figure 2—figure supplement 7*, PERMANOVA, p < 0.05). In addition to arthropod–host plant associations, we were able to detect interactions between arthropods. For example, abundances of the spruce gall midge *Piceacecis abietiperda* and its parasitoid, the chalcid wasp *Torymus* sp., were well correlated across all analyzed spruce sites (LM, p < 0.05). Both underwent coupled abundance cycles, with similar maxima every 6–8 years (*Figure 2C*).

The recovered community composition from ESB leaf homogenate samples was similar to hand-collected branch clipping samples (*Figure 2A, B*). Branch clipping recovered a larger diversity of spiders, a taxon which is exclusively found on leaf surfaces. In contrast, about 25% of the recovered taxa from ESB leaf powder likely inhabited the insides of leaves, e.g., gallers and miners (*Figure 2—figure supplement 6B*). Overall, arthropod DNA in leaf homogenates appears temporally very stable: Even freeze-dried leaf material that had been stored at room temperature for 8 years yielded surprisingly similar arthropod communities to ESB samples (*Figure 2A, B*). Besides analyzing diversity, we generated information on relative arthropod 18S rDNA copy number in relation to the corresponding plant 18S rDNA copy number by qPCR. Relative eDNA copy number should be a predictor for relative biomass. We designed a standardized qPCR assay based on lineage-specific blocking SNPs in the 18SrDNA gene (*Figure 3A*). We tested two primer combinations with 3'-blocking SNPs in (1) only the forward or (2) both forward and reverse primer sequences. The primer combination with mismatches in both forward and reverse primers led to a near complete suppression of non-arthropod amplification in all tested samples (5.41% vs. 44.49% on average) and was hence chosen for the qPCR experiment (*Figure 3A, B*). The qPCR assay accurately predicted changes in relative copy number of arthropod DNA on plant material across a 10,000-fold dilution series. Even when comparing taxonomically

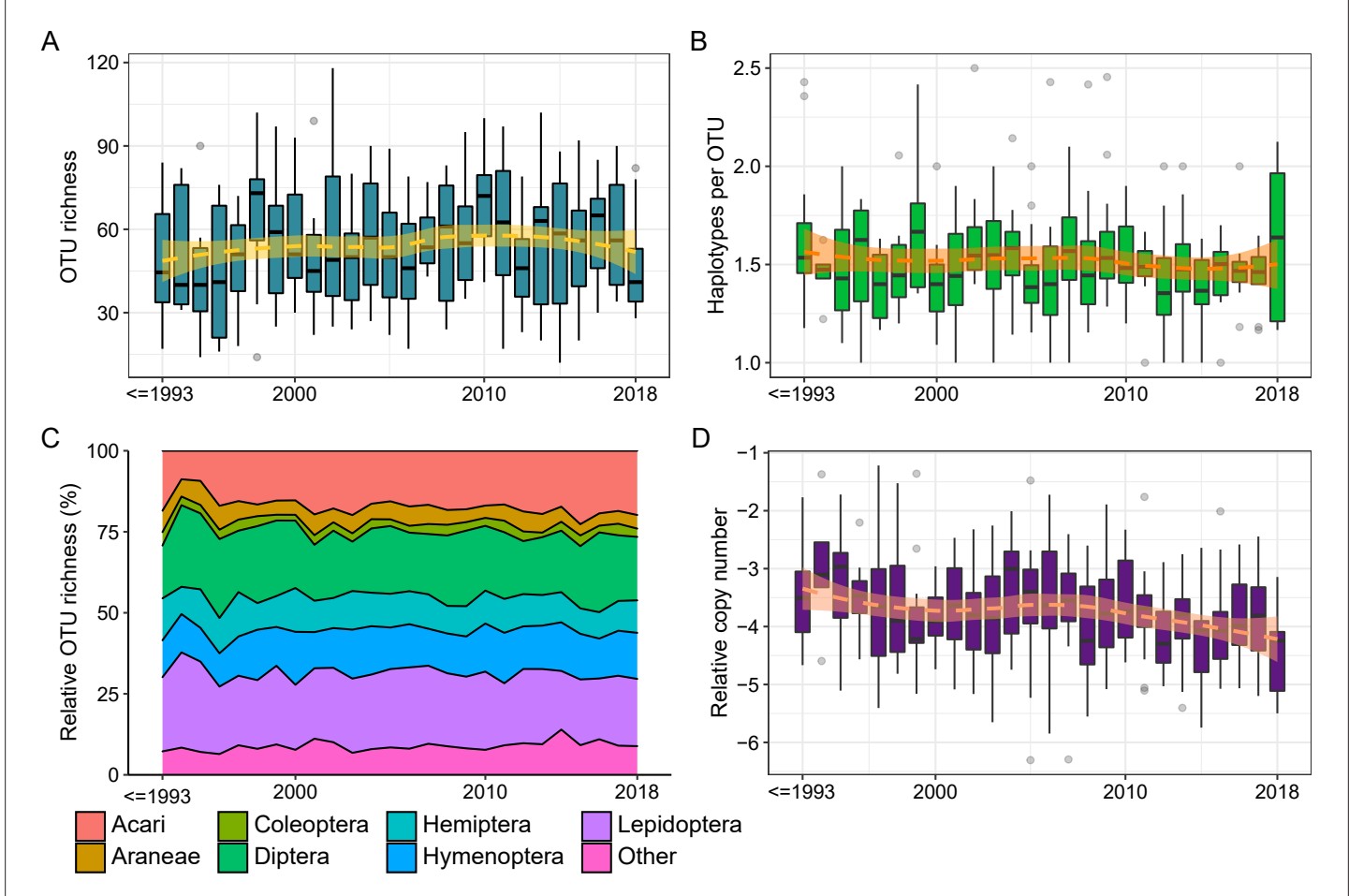

**Figure 4.** Temporal changes of diversity and copy number across all sites and samples (N = 312). (**A**) Arthropod OTU richness (representing α-diversity). (**B**) Haplotype richness within OTUs (representing genetic variation). (**C**) Relative OTU richness per order. (**D**) Relative copy number of arthropod DNA (representing biomass).

The online version of this article includes the following figure supplement(s) for figure 4:

**Figure supplement 1.** Metabarcoding-based diversity indices and quantitative PCR (qPCR)-based relative copy number of arthropod rDNA per land use type over three decades.

**Figure supplement 2.** Boxplots of OTU richness by decade and land use type for the six most speciose arthropod orders in our dataset.

**Figure supplement 3.** Arthropod diversity and relative copy number over time for all sites with time series longer than 10 years.

heterogeneous mock communities, very similar CT values were recovered (*Figure 3C*, *Figure 3— figure supplement 1*), highlighting the accuracy and wide applicability of our approach. Overall, we found a significant positive correlation of OTU richness and relative arthropod copy number in the ESB samples (LMM, $p < 0.05$; see Methods: '*Statistical analysis*'), supporting recent work suggesting a biomass–diversity relationship (*Hallmann et al., 2021*).

## Temporal changes of diversity, copy number, and species composition in canopy arthropod communities

Based on our time series data of archived ESB leaf samples, we tested the hypothesis that α-diversity (including intraspecific genetic diversity) and biomass (relative rDNA copy number) have undergone widespread temporal declines, particularly in areas of intensive land use. Our statistical analysis does not support previously reported widespread temporal α-diversity declines (*Figure 4A*, *Figure 2— figure supplement 5H, I*, LMM, $p > 0.05$), even when different land use types are analyzed separately (*Figure 4—figure supplement 1*, LMM, $p > 0.05$). Instead, warm summers and cold winters

were negatively associated with richness (LMM, p < 0.05). The temporal pattern of diversity was also largely independent of taxonomy: most orders did not show temporal trends when analyzed separately (*Figure 4C*; *Figure 4—figure supplement 2*). Exceptions include a significant loss of lepidopteran diversity, which is primarily driven by OTU loss at urban sites, and an overall increasing diversity of mites (LMM, p < 0.05). The overall temporally stable diversity is also visible at separate sites; a diversity decline across all orders was observed at only a single site (*Figure 4—figure supplement 3*). Similar to $\alpha$-diversity, we did not find widespread temporal declines of genetic diversity. Neither community-level zero radius OTU (zOTU) richness (which was well correlated to OTU richness, $R^2$ = 0.89) nor within-OTU haplotype richness declined significantly over time (*Figure 4B*; *Figure 4—figure supplement 1C, D*).

In contrast to the stable $\alpha$-diversity, relative copy number showed an overall decrease over time (*Figure 4D*, LMM, p < 0.05), suggesting that arthropod biomass may indeed be declining in woodlands (*Seibold et al., 2019*). The effect appears to be particularly driven by urban sites, coinciding with a loss of lepidopteran diversity (*Figure 4—figure supplements 1–3*). However, declines of arthropod DNA copy number are also visible in several agricultural and timber forest sites, particularly in the last 10 years of our time series (*Figure 4—figure supplement 1*, *Figure 4—figure supplement 3*).

We next explored temporal changes in abundance for 413 separate OTUs from a total of 19 sites. In line with our hypothesis (1), we predicted a majority of declining species. However, we found no significant difference between the average number of declining (6.94%) and increasing (10.04%) OTUs (*t*-test, p > 0.05, *Figure 5A*). With the exception of Acari, which showed an overrepresentation of increasing OTUs, declines and increases in OTU read abundance were independent of arthropod order and land use type (*Figure 5A, B*, Fisher's exact test p > 0.05). The observed replacement of about 15% of OTUs within sites translates into a significant temporal change of taxonomic $\beta$-diversity (*Figure 5C*, *Figure 2—figure supplement 5J and K*). We found a strong positive correlation of temporal distance and Jaccard dissimilarity for most sites (PERMANOVA, p < 0.05). Thus, species are continuously replaced in all land use types (*Figure 4—figure supplement 1F*). In the majority of analyzed sites, $\beta$-diversity did not show a correlation with differences in copy number (*Figure 5—figure supplement 2*).

While the community turnover did not affect local $\alpha$-diversity, we still observed associated losses of overall diversity. The first noteworthy pattern concerns a loss of temporal $\beta$-diversity within sites. $\beta$-Diversity between consecutive sampling years dropped significantly in many sites, particularly in beech forests (*Figure 5D*). Thus, diversity within sites is increasingly homogenized over time. We also found a significant decrease of $\beta$-diversity between sites for beech forests (*Figure 5E*). Our data suggest a loss of site-specific species and a gain of more widespread generalists, irrespective of land use. This pattern also emerges at the level of individual OTUs. Several novel colonizers spread rapidly in woodlands and showed similar abundance trends across various sites in parallel (*Figure 5—figure supplement 1*). The spatial and temporal change of $\beta$-diversity is illustrated by an NMDS plot of arthropod communities from two beech forests in National Parks, the Harz and the Bavarian Forest (*Figure 5F*). While the two sites are well separated by the first NMDS axis, the second axis shows a pronounced temporal turnover of communities. In the past decades, this turnover has led to temporally more predictable communities within sites and increasingly similar communities between the two national parks.

## Discussion

Here, we show that DNA from archived leaf material provides a robust source of data to reconstruct temporal community change across the arthropod tree of life. Leaf samples should also cover a broad phenological window: Adults of many insect species are only active during a short time period of the year, but their larvae spend the whole year on their host plant (*Gagné and Graney, 2014*). A Malaise trap will miss these taxa most times of the year, while an eDNA approach should detect the larvae throughout. As sampled leaves make up the habitat and often food source of arthropods, it is also possible to infer the exposure of arthropod communities to chemical pollution by analyzing chemicals in the leaf material. This is of critical importance, as pesticide use has often been invoked as a driver of insect decline (*Goulson et al., 2015*; *Siviter et al., 2021*). The high temporal stability of arthropod DNA in 8-year-old dried leaf samples also suggests the utility of other plant archives, for example, herbaria, for arthropod eDNA. However, for such less standardized and less well-preserved sample

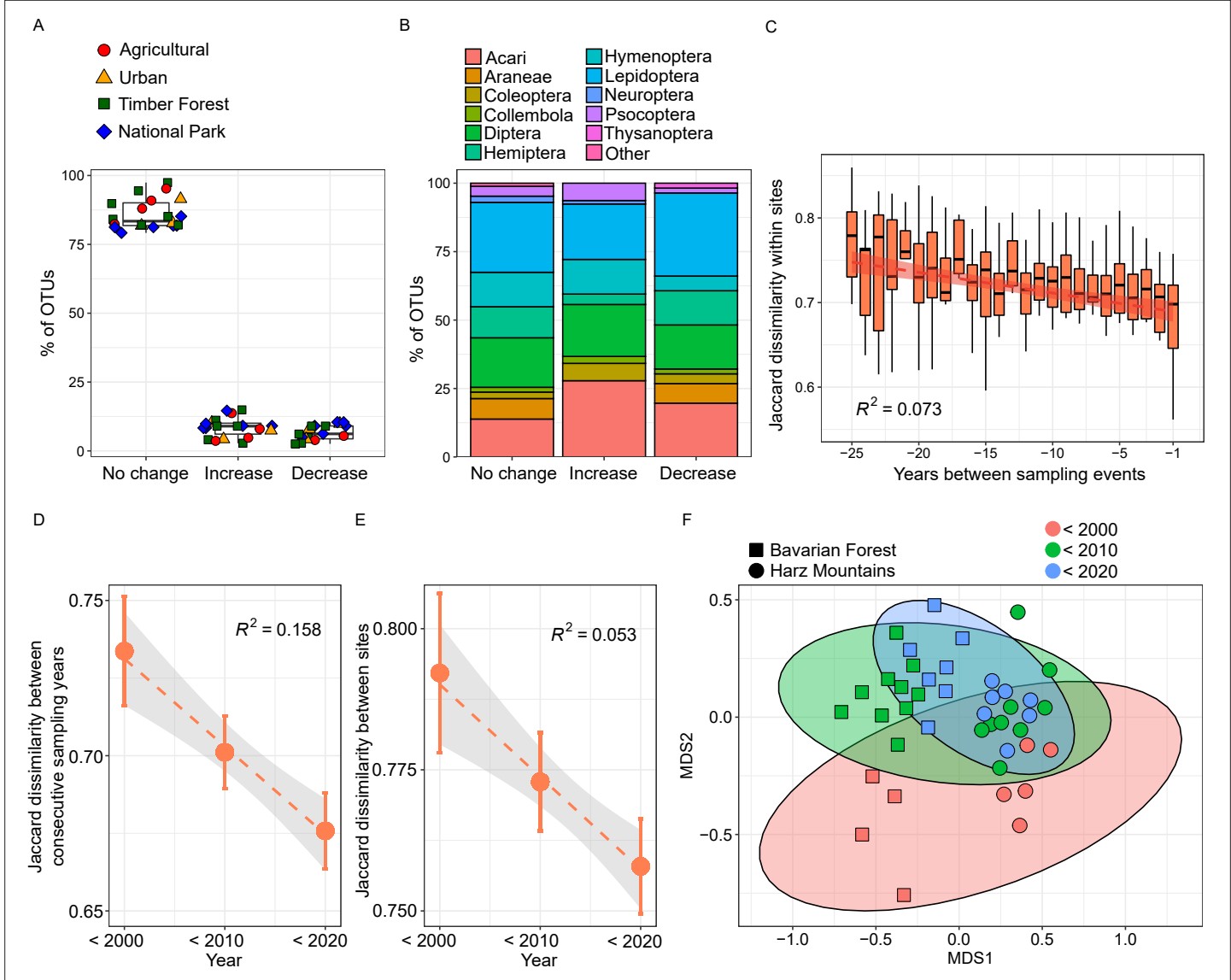

**Figure 5.** Temporal changes of species composition and β-diversity within and between sampling sites. (**A**) Boxplots of the proportion of 413 separate OTUs that significantly increased, decreased, or did not show temporal abundance changes. The colored symbols overlaying the boxplots represent land use type for each site. (**B**) Stacked barplot showing the recovered order composition of the same three categories of OTUs. Orders amounting to less than 1% of the total OTU number are merged as 'Other'. (**C**) Within-site Jaccard dissimilarity as a function of number of years between sampling events, showing a pronounced taxonomic turnover (N = 312). (**D**) Jaccard dissimilarity between consecutive sampling years in each decade of sampling (N = 15, 44 & 39), calculated within sites for beech forests. We find a loss of temporal β-diversity over time. Points indicate means and error bars show 95% confidence interval. (**E**) Jaccard dissimilarity between sites for the same samples as in D. We find a loss of average between-site β-diversity, that is spatial homogenization. (**F**) NMDS plot showing community dissimilarity within and between Bavarian Forest and Harz Mountains national parks over three decades (<2000: sampled before 2000; <2010: sampled between 2000 and 2009; <2020: sampled between 2010 and 2018).

The online version of this article includes the following figure supplement(s) for figure 5:

**Figure supplement 1.** Temporal change in relative read abundance for five exemplary arthropod OTUs at all beech and poplar sites and abundance change for three OTUs at two sites each in Germany.

**Figure supplement 2.** Correlation of dissimilarity in 18S rDNA copy number and β-diversity for all sites with time series longer than 10 years.

types, careful evaluation of cross-contamination or chemical DNA modification (*Orlando et al., 2021*), which could inflate the recovered diversity, may be warranted.

We here analyze a unique leaf archive and provide unprecedented insights into arthropod community change in the tree canopy, an ecosystem known for its high and often cryptic diversity (*Nakamura*

*et al., 2017*). Our results do not confirm the hypothesis of widespread losses of $\alpha$-diversity. Initial reports of insect declines originate mainly from grassland ecosystems that have undergone massive changes in land use (*Hallmann et al., 2017*). Central European canopy communities may be less affected by such change. Interestingly, the only sites that showed declines of richness were agricultural and urban sites, suggesting that land use may at least locally affect neighboring canopy communities (*Hallmann et al., 2014*). Our data also suggest negative effects of warm summers on richness. Climate warming has recently been suggested to act in conjunction with land use change to drive insect declines (*Outhwaite et al., 2022*).

Instead of declining richness, we detected DNA copy number declines in all land use types, suggesting that overall biomass may indeed be in decline. Extinction is the endpoint of a long trajectory of decline and increasingly affects common species (*Collen et al., 2011*), hence biomass decline could foreshadow future biodiversity loss. Alternatively, the dropping copy number may reflect a taxonomic turnover of species with either different eDNA shedding rates or different rDNA copy numbers in the different communities. However, if the latter were true, an association of copy number changes and turnover would be expected, which we did not find in our data.

Instead of widespread losses of $\alpha$-diversity, we indeed found pronounced taxonomic turnover in nearly all communities (*Dornelas et al., 2014*), supporting our second hypothesis. While resident species are continuously lost, they are mostly replaced by novel colonizers. This turnover can result in biotic homogenization across space and time. Less interannual variation of the occurrence of taxa within sites and reduced spatial variation of their occurrence between sites both cause a decline in overall $\beta$-diversity. Biotic homogenization is often associated with intensification of land use (*Karp et al., 2012*; *Gossner et al., 2016*) and landscape simplification (*Holland et al., 2005*). However, the pattern we observed affected all land use types equally. The universality of these changes suggests that neither site- nor taxon-specific factors are responsible. Possible explanations include factors that act at a larger scale, such as climate change-induced range shifts (*Marta et al., 2021*), nitrogen deposition (*Gámez-Virués et al., 2015*), and the introduction of invasive species (*Soroye et al., 2020*; *Lister and Garcia, 2018*). Given that our leaf samples recover a fairly broad phenological window, the alternative explanation that the observed community-wide turnover pattern may have resulted from shifting phenologies (*Cohen et al., 2018*) is unlikely. The gradual replacement of species also suggests that we are not yet observing ecosystems reaching tipping points (*Barnosky et al., 2012*).

In summary, our work shows the great importance of standardized time series data to accurately reveal biodiversity change in the Anthropocene (*Thomsen et al., 2016*) across space and time, beyond the decline of $\alpha$-diversity and biomass (*Dornelas et al., 2014*). Taxonomic replacement and biotic homogenization, even in seemingly pristine habitats such as national parks, signify an important and hitherto insufficiently recognized facet of the current insect crisis.

## Materials and methods
### Samples and metadata used in this study
Tree samples of the German Environmental Specimen Bank – standardized time series samples stored at ultra-low temperatures
We used a total of 312 leaf samples of four common German tree species: the European Beech *Fagus sylvatica* (98 samples), the Lombardy Poplar *Populus nigra 'italica'* (65 samples), the Norway Spruce *Picea abies* (123 samples), and the Scots Pine *Pinus sylvestris* (26 samples). The samples have been collected annually or biannually by the German Environmental Specimen Bank (ESB) since the 1980s and serve as indicators for aerial pollutants (*Schulze et al., 2007*). A total of 24 sampling sites were included, covering sampling periods of up to 31 years and representing four land use types of varying degrees of anthropogenic disturbance (*Figure 1*). These include natural climax forest ecosystems in core zones of national parks (six sites, National Park), forests commercially used for timber (six sites, Timber Forest), tree stands in close proximity to agricultural fields (six sites, Agricultural), and trees in urban parks (three sites, Urban). The sites were initially chosen to represent their land use type permanently for long-term monitoring, and the corresponding land use categories have mostly remained temporally stable.

ESB samples are collected and processed according to a highly standardized protocol at the same time every year. Sampling events between different years of the time series usually do not differ by

more than 2 weeks. All used equipment is sterilized before field work by several washes and heat treatments (*Tarricone et al., 2018a*; *Tarricone et al., 2018b*; *Klein et al., 2018*). A defined amount of leaf material (>1.100 g) is collected from a defined number of trees (15 at most sites) from each site and from 4 branches from each tree. The branches are distributed equally spaced in the outer crown area of the tree. The amount sampled translates to several thousand leaves from each site, which should suffice to saturate the recovered arthropod diversity. For a subset of samples, biometric analysis is performed, for example the weight of individual leaves and general condition of the tree are noted. Leaf weight has not changed over the time series at most sites. The sampled leaves are intended to represent the exact natural state of the tree. They are not washed or altered in any way before processing, and eDNA traces, and small arthropods on the leaves' surfaces, as well as from galls and leaf mines, are included in the sample. Each sample is stored on liquid nitrogen immediately after collection and ground to a powder with an average diameter of 200 µm using a cryomill. The cryomill is thoroughly cleaned between samples to prevent cross-contamination. The resulting homogenates are then stored for long term on liquid nitrogen (*Rüdel et al., 2009*; *Rüdel et al., 2015*). The cold chain is not interrupted after collection and during processing, ensuring optimal preservation of nucleic acids in the samples. The homogenization of the sample also guarantees a thorough mixing, resulting in equal distribution of environmental chemicals and probably eDNA in the sample. Previous work suggests that very small subsamples of the homogenate suffice to detect even trace amounts of environmental chemicals in the sample (*Gámez-Virués et al., 2015*). eDNA from leaf surfaces may be affected by weather conditions before sampling, for example, washed away by heavy rain or damaged by strong UV exposure (*Valentin et al., 2020*). However, leaves are only collected dry by the ESB, that is, not immediately after rain. The date of the most recent rainfall is noted for each ESB sampling event, allowing us to explore the effects of recent weather conditions on the recovered arthropod communities.

## The utility of plant material stored at room temperature to recover arthropod DNA

ESB samples are stored under optimal conditions for nucleic acid preservation. By contrast, most archived leaf samples are stored at room temperature, for example dried leaf material in herbaria. To test the general suitability of archived leaf material for arthropod community analysis, we included 25 additional beech leaf samples, each consisting of 100 leaves from a total of 5 sites. The samples were freeze-dried and then ground to a fine powder by bead beating. The resulting powder was comparable to our ESB homogenates, but unlike them, it was stored at room temperature for 6–8 years.

## Hand-collected branch clipping samples to explore the accuracy of leaf-derived arthropod DNA

To evaluate the performance of our leaf DNA-based protocol in comparison to commonly used sampling methods, we generated a branch clipping dataset from five beech stands close to Trier University. Branch clipping is a widely used method to collect arthropods residing on leaf surfaces in trees (*Delvare, 1997*) and thus the best comparable traditional methodology to our protocol. We sampled five trees per site and collected ten branches of about 40 cm length from each tree. The branches were clipped off, stored in plastic bags and then brought to the laboratory. Here, arthropods were manually collected from each sample. All collected arthropod specimens were pooled by tree and stored for later DNA extraction in 99% ethanol.

## Climate data

We downloaded monthly climate data for all study sites from the German Climate Center distributed as a raster dataset interpolated from the surrounding weather stations by the German Meteorological Service (Deutscher Wetterdienst – DWD). We collected data for average annual temperature and rainfall as well as summer and winter temperature and rainfall.

## Measurement of pesticide content from archived leaf material

The ESB sampling was historically set up as a tool for pollution assessment (*Schulze et al., 2007*), and the samples are therefore stored to preserve any possible pollutant. Because these leaves serve as a habitat for the associated arthropods, such well-preserved material allows us to explore levels of

chemical pollution occurring directly within the arthropods' environment. Our samples were screened for pesticides and persistent organic pollutants with a modified QuEChERS approach (*Löbbert et al., 2021*). 2.0 g of sample material were extracted with acetonitrile (10 ml) and ultrapure water (10 ml), followed by a salting-out step using magnesium sulfate, sodium chloride and a citrate buffer (6.5 g; 8:2:3 [wt/wt]). After a dispersive solid-phase extraction cleanup step with magnesium sulfate, PSA (primary-secondary amine), and GCB (graphitized carbon black) (182.5 mg; 300:50:15 [wt/wt]), the supernatant was analyzed with a sensitive gas chromatography–tandem mass spectrometry (GC–MS/MS) instrument. All samples were analyzed for 208 GC-amenable compounds of different pesticide and pollutant classes, including pyrethroids, organochlorine and organophosphate pesticides, and polychlorinated biphenyls.

## Molecular processing
### DNA isolation
We developed a highly standardized protocol for the analysis of leaf-associated arthropod community DNA. We optimized various protocol steps to ensure the reliability and reproducibility of our data. We first explored the effect of DNA extraction on recovered diversity.

As mentioned above, the cryo-homogenization of ESB samples ensures a very homogeneous distribution of even trace amounts of chemicals in the sample. This should also hold true for DNA. Hence small subsamples of large homogenate samples should suffice for analysis. To test this hypothesis, we first performed a weight series extraction from 50, 100, 200, 400, 800, and 1600 mg of homogenate with several replicates for each weight. Additional extraction replicates of 16 beech samples at 200 mg were also included. This analysis confirms the pronounced homogenization of the samples, with 200 mg of homogenate sufficing to accurately recover $\alpha$- and $\beta$-diversity (*Figure 2—figure supplement 1A, B*).

A single DNA extract was made from each ESB and freeze-dried sample, using the Puregene Tissue Kit according to the manufacturer's protocols (Qiagen, Hilden, Germany). All samples were processed under a clean bench and kept over liquid nitrogen during processing to prevent thawing. Samples were transferred using a 1000-µl pipette with cutoff tips. The resulting wide bore tips were used to drill out cores of defined sizes from the leaf powder. To remove undesired coprecipitates, we performed another round of purification for each sample, using the Puregene kit following the manufacturer's protocol. The hand-collected arthropod specimens from our branch clipping were pulverized in a Qiagen Tissuelyzer at 200 Hz for 2 min using new 5-mm stainless steel beads, and DNA was extracted from the pulverized samples using the Puregene kit as described in *de Kerdrel et al., 2020*. Branch clipping and leaf samples were processed in separate batches and using separate reagents to avoid possible carryover between sample types.

## Primer choice, PCR amplification, and sequencing
As the standard DNA barcode marker for arthropods (*Andújar et al., 2018*), the mitochondrial COI gene offers the best taxonomic identification of German arthropod species. We thus selected COI for our metabarcoding analysis. We tested several primer pairs to optimize recovery of arthropod DNA from the leaf homogenates (*Gibson et al., 2014*; *Leray et al., 2013*; *Jusino et al., 2019*). Leaf homogenates are dominated by plant DNA, with arthropod eDNA only present in trace amounts. The majority of commonly used arthropod metabarcoding primers are very degenerate and will readily amplify mitochondrial COI of plants. Thus, for such degenerate primers, the vast majority of recovered reads will belong to the plant. We found an ideal tradeoff between suppressing plant amplification while still recovering a taxonomically broad arthropod community in the primer pair ZBJ-ArtF1c/ZBJ-ArtR2c (*Zeale et al., 2011*). While this primer pair is known to have taxonomic biases for certain arthropod groups (*Piñol et al., 2015*), it is still widely used as an efficient and reliable marker for community analysis (*Thomsen and Sigsgaard, 2019*; *Eitzinger et al., 2021*). Recently, we designed a novel and highly degenerate primer pair by modifying two degenerate metabarcoding primers (*Gibson et al., 2014*; *Leray et al., 2013*), which allows the suppression of plant amplification (*NoPlantF_270/mlCOIintR_W*; *Supplementary file 1*; *Krehenwinkel et al., 2022*). To ensure the reproducibility of the diversity patterns recovered from our original ZBJ-ArtF1c/ZBJ-ArtR2c dataset, we additionally processed eleven complete ESB time series (174 samples) for this novel primer pair and compared results for species composition, $\alpha$- and $\beta$-diversity. This analysis supports very similar

patterns for temporal species abundance trends, as well as $\alpha$- or $\beta$-diversity for both primer pairs (**Figure 2—figure supplement 5**).

All PCRs were run with 1 µl of DNA in 10 µl volumes, using the Qiagen Multiplex PCR kit according to the manufacturer's protocol and with 35 cycles and an annealing temperature of 46°C. A subsequent indexing PCR of five cycles at an annealing temperature of 55°C served to attach sequencing adapters and 8-bp dual indexes (all with a minimum 2 bp difference) to each sample (using the layout described in **Lange et al., 2014**). We had previously tested the effect of DNA extraction and PCR replicates, which showed well correlated and reproducible OTU composition ($R^2_{extract1vs2}$ = 0.90, $R^2_{PCR1vs2}$ = 0.97, LM p < 0.05), as well as $\alpha$-diversity patterns ($R^2_{extract1vs2}$ = 0.93, $R^2_{PCR1vs2}$ = 0.90, LM p < 0.05). PCR and extraction replicates also recovered a significantly lower $\beta$-diversity than within- and between-site comparisons ($\beta_{PCR1vs2}$ = 0.14; $\beta_{extract1vs2}$ = 0.19; $\beta_{within\ site}$ = 0.62; Pairwise Wilcoxon Test, p < 0.05) (**Figure 2—figure supplement 1**). Considering the similar recovered diversity patterns for PCR and extraction replicates and the observed saturation of diversity with single 200 mg homogenate extractions, we did not perform extraction replicates, but ran all PCRs in duplicate as technical replicates. To assure that dual replicates suffice to recover diversity patterns, we also added a PCR triplicate for two time series of beech samples (42 samples in total) and sequenced each of these samples to 78,000 reads on average. Patterns of diversity were highly correlated between duplicate and triplicate datasets ($R^2$ = 0.95). The final libraries were quantified on a 1.5% agarose gel and pooled in approximately equal abundances based on gel band intensity. The final pooled sample was cleaned using 1× Ampure Beads XP (Beckmann-Coulter, Brea, CA, USA) and then sequenced on an Illumina MiSeq (Illumina, San Diego, CA, USA) using several V2 kits with 300 cycles at the Max Planck Institute for Evolutionary Biology in Plön, Germany. Branch clipping samples were amplified and sequenced separately using the above protocol. Negative control PCRs and blank extraction PCRs were run alongside all experiments and sequenced as well, to explore the effect of possible cross-contamination or index carryover between samples.

## Test for DNA carryover in the cryomill

The sample processing pipeline of the ESB is laid out to be sterile and entirely avoid cross-contamination between samples. To test the efficiency of these protocols for eDNA sampling, we included a test on the possibility of carryover in the cryomill. Using the milling schedule of tree samples from 2015 to 2018, we compared the $\beta$-diversity between tree samples that were processed in the cryomill consecutively. Assuming an eDNA carryover takes place, the $\beta$-diversity should be significantly reduced compared to samples which are processed in different years. We did not find an effect of processing order in the cryomill on beta diversity for 18 within- and between-tree species comparisons (**Figure 2—figure supplement 3**). We also explored the effect of single species carryover in the cryomill. This was done using samples of different tree species, which were processed consecutively in the cryomill. We compared the read abundances of the 10 most abundant monophagous species with that found in the consecutively processed sample of a different tree species. The comparisons were done for one poplar and beech sample as well as one pine and spruce sample. To ensure even minor carryover would be detected, we sequenced all samples to a high depth of 78,000 reads on average. Yet, no signal of carryover was observed (**Supplementary file 3**).

## Sequence processing

Reads were demultiplexed by dual indexes using CASAVA (Illumina, San Diego, CA, USA) allowing no mismatches in indexes. Demultiplexed reads were merged using PEAR (**Zhang et al., 2014**) with a minimum overlap of 50 and a minimum quality of 20. The merged reads were then quality filtered for a minimum of 90% of bases >Q30 and transformed to fasta files using FastX Toolkit (**Gordon and Hannon, 2010**). Primer sequences were trimmed off using *sed* in UNIX, with degenerate sites allowed to vary and only retaining sequences beginning with the forward and ending with the reverse primer. The reads were then dereplicated using USEARCH (**Edgar, 2010**). The dereplicated sequences were clustered into zero radius OTUs (hereafter zOTUs) using the *unoise3* command (**Edgar, 2016**) and 3% radius OTUs using the *cluster_otus* command in USEARCH with a minimum coverage of 8 and a minimum occurrence of three reads in a sample. Chimeras were removed de novo during OTU clustering. All resulting sequences were translated in MEGA (**Takahara et al., 2012**) and only those with intact reading frames were retained. To assign taxonomic identity to the zOTU sequences, we used

BLASTn (*Altschul et al., 1990*) against the complete NCBI nucleotide database (downloaded February 2021) and kept the top 10 hits. Sequences were identified to the lowest possible taxonomic level, with a minimum of 98% similarity to classify them as species. All non-arthropod sequences were removed. We then built Maximum Likelihood phylogenies from alignments of the zOTU sequences for all recovered arthropod orders separately using RaxML (*Stamatakis, 2014*). These phylogenies were used to perform another clustering analysis using ptp (*Zhang et al., 2013*) to generate OTUs from the data. Due to the well-developed German Barcode of Life database (*Geiger et al., 2017*), actual species identity can be reliably inferred by database comparisons for many arthropod groups. 3% radius OTUs often oversplit species, for example several 3% radius OTUs comprised one actual species. The ptp clustering often merged several 3% radius OTUs, but came closest to the actual species assignments by BLAST. Moreover, the recovered diversity values for 3% OTUs, zOTUs and ptp-based OTUs were well correlated ($R^2 = 0.90$). We thus proceeded to use ptp-based OTUs (hereafter referred to as OTUs) for subsequent analysis on taxonomic diversity, as it should best approximate actual species diversity. zOTUs represent individual haplotypes in the dataset and were used as an indicator of genetic diversity. Using the taxonomic assignments, we estimated which taxonomic groups were particularly well represented in our data. Each tree species likely harbors a unique arthropod community with numerous monophagous species, a majority of which should be recovered by a broadly applicable molecular method. Where possible, we performed a finer scale ecological assessment for the recovered taxa, classifying them by trophic ecology and expected position on the outside or inside of the leaf. For example, mining taxa would likely be recovered from the inside of the leaf, while other taxa likely reside on the leaf's surface.

## Detection of relative arthropod DNA copy number using qPCR

Initial reports on insect decline were entirely based on biomass (*Hallmann et al., 2017*). Biomass, however, does not necessarily predict diversity (*Gough et al., 1994*). We therefore aimed to generate information not only on diversity, but also on relative biomass of arthropods in tree canopies. Previous eDNA studies show that DNA copy number is correlated with the biomass of a target taxon (*Takahara et al., 2012*), making qPCR a possible approach for biomass estimation. We developed a qPCR protocol to detect relative abundance of arthropod DNA copy number in leaf samples, using the plant DNA copy number as an internal reference for quantification. We used the nuclear 18SrRNA gene (hereafter 18S). Although 18S can show interspecific copy number variation, it provides relatively good approximations of actual taxon abundances in amplicon assays (*Krehenwinkel et al., 2017*; *Krehenwinkel et al., 2019b*). Primer pairs targeting plants and arthropods were designed to meet the following criteria: (1) Identical PCR fragments should be amplified for plants and arthropods so that PCR for both taxa will perform similarly. (2) The arthropod-specific primer should not amplify plants, and vice versa. (3) Fungi should be excluded from amplification, as DNA of fungal endophytes is probably at least as abundant in leaf samples as arthropod eDNA. (4) The primers should target conserved regions in order to amplify a broad spectrum of plants or arthropods. We used diagnostic SNPs at each primer's 3'-end to achieve the lineage specificity (*Krehenwinkel et al., 2019a*).

Two possible qPCR primer pairs were designed, one targeting a 172 bp and the other a 176 bp fragment of 18S (*Supplementary file 1*). The first primer pair contained a 3'-AA-mismatch discriminating arthropods from plants and fungi in the forward primer and a 3'-TT-mismatch discriminating plants from arthropods in the reverse primer, while the second pair had the same 3'-AA-mismatch in the forward primer but no reverse primer mismatch (*Figure 3A*). To test the lineage specificity of both primer pairs, we performed an amplicon sequencing experiment with the arthropod-specific primers. Four samples from each of the four tree species were amplified with both primer pairs, then indexed, pooled, sequenced, and processed as described above. All reads were clustered into 3% radius OTUs. Taxonomy was assigned to the OTUs, and an OTU table was built, as described above for the ESB sample metabarcoding experiment. The proportion of arthropod reads was then estimated for each sample and primer pair. The first primer combination (3'-mismatch in both forward and reverse primers) led to a near complete suppression of plant and fungal amplification in all tested samples and was therefore used for the qPCR (*Figure 3A, B*).

To account for the low quantity of arthropod DNA in relation to plant DNA, we used a nested qPCR assay, with a high accuracy for low DNA copy numbers (*Tran et al., 2014*). The sample was first amplified in a regular PCR with 15 cycles using the Qiagen Multiplex PCR kit. Two separate PCRs were

run: one using the arthropod primers with an undiluted DNA extract as template, and the other using the plant primers with a 1:100 dilution of the DNA extract. The primers included a 33-bp forward and 34-bp reverse tail, based on Illumina TruSeq libraries, which were complementary to sequences in the qPCR primers. After being cleaned of residual primers with 1× AMPure beads XP, the products of the first PCR were used as template in the qPCR. qPCR was run with the Power SYBR Green Mastermix (Fisher Scientific, Waltham, MA, USA) on an ABI StepOnePlus Real-Time PCR System (Applied Biosystems, Foster City, CA, USA) according to the manufacturer's protocol, using 35 cycles and an annealing temperature of 55°C. All reactions were run in triplicate and the average of the three CT values used for analysis. CT values showed high reproducibility between triplicate PCRs ($SD_{plant}$ = 0.13; $SD_{arthropod}$ = 0.10). Non-template controls were run alongside all qPCRs to rule out contamination.

The qPCR efficiency was estimated for the plant and arthropod-specific marker using two ESB tree samples. A 10,000-fold dilution series was used for efficiency estimation. This assay was very stringent, as it corresponds to a dilution of the naturally occurring arthropod eDNA in a plant sample by 10,000. Both assays showed a high efficiency across the dilution series ($E_{Plant}$ = 94.77%, $E_{Arthropod}$ = 99.73%). The 1:10000 dilution ($CT_{plant\ 1.1000}$ = 15.4; $CT_{insect\ 1.1000}$ = 27.2) is far less than the actual amount of insect DNA in an ESB sample (average $CT_{insect}$ = 21.3; average $CT_{plant}$ = 15.7), supporting the reliability of our experiment.

To estimate the accuracy of relative arthropod DNA copy detection across diverse arthropod communities, we also performed a spike-in assay, in which a dilution series (10,000-fold) of arthropod mock community DNA was added to a leaf extract and analyzed using qPCR. Seven mock communities were prepared, each containing varying amounts of DNA from 13 arthropod species representing 13 different orders (*Figure 3—figure supplement 1*). The relative copy number of arthropod DNA in relation to plant DNA was estimated using the Delta CT Method (*Schmittgen and Livak, 2008*). The optimized qPCR protocol was then used to quantify the relative DNA copy number of arthropods in all 312 ESB leaf samples.

## Statistical analysis

Using USEARCH, an OTU table was built including all samples with the taxonomically annotated zOTU sequences as reference. A species-level OTU table was then generated by merging the zOTUs into their respective *ptp*-based OTU clusters.

The negative control samples were mostly free of arthropod sequences. We found 1.88 arthropod reads per control on average (0–5 reads per control). The recovered reads belonged to taxa that were highly abundant in one of the analyzed tree species, suggesting minor carryover during PCR or sequencing. Based on the negative control samples, we removed all entries in the OTU table with fewer than three reads to counter this possible carryover.

We used two approaches to rarefy our OTU table. First, using rarefaction analysis in vegan (*Oksanen et al., 2013*) in R (v 4.1.0) (*Team RS, 2015*), we explored saturation of diversity. Based on this analysis, 5000 reads were randomly sampled for each of the two PCR duplicates using GUniFrac (*Chen et al., 2018*), and the duplicates were merged into a final sample of 10,000 reads (*Figure 2—figure supplement 4*) after the replicates were checked for reproducible patterns of species composition, $\alpha$- and $\beta$-diversity. To ensure that undersampling did not affect our results, we performed an additional analysis with the unrarefied dataset, which yielded an average coverage of 21,676 reads per sample. A second rarefaction was informed by the relative copy number of arthropod DNA recovered from the tree samples with our qPCR assay. Assuming the copy number reflects biomass, we sampled read numbers proportional to the specific relative copy number for each sample. The copy number-informed and unrarefied datasets showed highly correlated diversity patterns with the dataset rarefied to 10,000 reads ($R^2 \geq 0.91$).

Taxonomic $\alpha$- and $\beta$-diversity were calculated in vegan in R. Quantitative biodiversity assessments by metabarcoding at the community level are likely biased (*Krehenwinkel et al., 2017*). We therefore limited our assessments of $\alpha$-diversity to richness and $\beta$-diversity to binary dissimilarity.

We also measured temporal abundance changes of single OTUs within sites. Within OTUs, temporal changes in read abundance at a site should reflect the relative abundance with reasonable accuracy (*Krehenwinkel et al., 2017*). Only sites spanning a minimum time series of 10 years were included, and we only used species that occurred in at least three sampling events for a particular site and for which at least 100 reads were recovered. This filtering served to exclude rare species, which imitate

abundance increases or declines by randomly occurring early or late in the time series. To account for likely fluctuations in abundance, we used the log + 1 of read abundance. Significant increases or declines of abundance over time were estimated for each OTU and site using non-parametric Spearman correlation in R. Both the qPCR-informed and the rarefied datasets were used to calculate species abundance changes.

As mentioned above, zOTUs represent individual haplotypes and thus genetic variation within species. Using the zOTU data, we calculated the haplotype (zOTU) richness within each individual OTU as a complementary measure for genetic variation. A decline in biodiversity could manifest itself in an overall loss of species, which should be detectable at the OTU level. Alternatively, biodiversity decline could initially only affect genetic variation within species, for example, be the result of declining population sizes without actual extinctions. This should be detectable by losses of overall zOTU diversity and zOTU richness within single OTUs. To derive within-OTU genetic diversity, we identified OTUs that consisted of more than one zOTU. zOTU variation could be affected by low abundance sequence noise. Hence, we used the same filtering criteria as described above for the species abundance change. Moreover, we only included zOTUs in our calculations that were present in both technical replicates of a sample. The richness of the remaining zOTUs within each of these OTUs was then calculated.

Arthropods are an ecologically very heterogeneous taxon, with different groups showing very different life histories and possibly responses to ecosystem change. To account for this heterogeneity, we calculated $\alpha$- and $\beta$-diversity metrics for the complete arthropod dataset, as well as the 10 most common arthropod orders in the dataset. Using NMDS ($k$ = 2, 500 replications, Jaccard dissimilarity) in vegan, we visualized differentiation of the recovered arthropod communities. We then tested for effects of tree species, sampling year, sampling site, land use type, weather before sampling, amount of leaf material in a sample, climatic variables and detected pesticide load on $\alpha$- and $\beta$-diversity. Also, hand-collected branch clipping samples, as well as the freeze-dried samples, were compared with the ESB samples for their recovered arthropod community composition and diversity.

Factors contributing to $\beta$-diversity were evaluated using a PERMANOVA in *vegan*. To evaluate an association of community turnover and copy number variation (e.g., biomass), we also explored patterns of association between these two variables for each site (***Figure 5—figure supplement 2***). Statistical analysis for temporal changes of $\alpha$-diversity and relative copy number were performed using the *nlme* (v 3.1-159; 2022) (***Bates et al., 2014***) package in R. LMMs were applied to analyze the statistical importance of involved fixed and random effects. Temperature (annual, summer, and winter temperatures), corresponding rainfall data, year, and land use type were treated as fixed effects. Site ID was included as random effect. The Akaike information criterion was used in stepwise regression to identify the final models. A continuous-time first-order autocorrelation model term (corCAR1) was included in the *lme* function to account for serial autocorrelation. The dominant predictor variable is the tree species, which always contributes most to the marginal $R^2$ in all models. Arthropod DNA copy number (marginal $R^2$ = 0.26) showed negative associations with time (p < 0.001) and winter temperatures (p = 0.038). None of the other diversity metrics (OTU and zOTU richness, copy number-corrected richness, saturated richness, and genetic diversity) showed an association with time. However, all richness values were positively correlated to winter temperatures and negatively to summer temperatures (marginal $R^2$ = 0.45–0.48, p < 0.05). Genetic diversity (marginal $R^2$ = 0.34) was correlated to winter rainfall (p < 0.001).

## Acknowledgements

We thank Karin Fischer for assistance with lab work. The German Environment Agency made the leaf samples available. Thanks to Diethard Tautz, Natalie Graham, and Rosemary Gillespie for critical revision of an earlier version of the manuscript. Frank Thomas, Dorothee Krieger, and Bernhard Backes provided access to the dried leaf homogenate we used here. Andrea Koerner helped in acquiring the leaf samples from the ESB. SW was funded by a PhD fellowship of the German Federal Environmental Foundation (DBU).

## Additional information

### Funding

| Funder | Grant reference number | Author |
|---|---|---|
| Deutsche Bundesstiftung Umwelt | | Henrik Krehenwinkel |

The funders had no role in study design, data collection and interpretation, or the decision to submit the work for publication.

### Author contributions

Henrik Krehenwinkel, Conceptualization, Formal analysis, Supervision, Investigation, Methodology, Writing – original draft, Project administration, Writing – review and editing; Sven Weber, Formal analysis, Visualization, Methodology, Writing – original draft, Writing – review and editing; Rieke Broekmann, Anja Melcher, Investigation; Julian Hans, Investigation, Methodology, Writing – review and editing; Rüdiger Wolf, Data curation, Methodology; Axel Hochkirch, Jan Koschorreck, Investigation, Writing – review and editing; Susan Rachel Kennedy, Supervision, Investigation, Visualization, Methodology, Writing – original draft, Writing – review and editing; Sven Künzel, Diana Teubner, Resources, Methodology; Christoph Müller, Rebecca Retzlaff, Sonja Schanzer, Investigation, Methodology; Roland Klein, Conceptualization, Resources, Investigation; Martin Paulus, Conceptualization, Resources, Funding acquisition, Investigation; Thomas Udelhoven, Formal analysis, Investigation, Methodology; Michael Veith, Conceptualization, Resources, Funding acquisition, Writing – review and editing

### Author ORCIDs

Henrik Krehenwinkel  http://orcid.org/0000-0001-5069-8601
Rüdiger Wolf  http://orcid.org/0000-0002-8144-5954
Susan Rachel Kennedy  http://orcid.org/0000-0002-1616-3985
Roland Klein  http://orcid.org/0000-0001-8735-0393
Michael Veith  http://orcid.org/0000-0002-7530-4856

### Decision letter and Author response

Decision letter https://doi.org/10.7554/eLife.78521.sa1
Author response https://doi.org/10.7554/eLife.78521.sa2

## Additional files

### Supplementary files

• Supplementary file 1. Primers used in this study.

• Supplementary file 2. OTU table with metadata and quantitative PCR (qPCR) results.

• Supplementary file 3. Contamination check in the cryomill. The table shows the taxonomic annotation for the 10 most abundant host-specific OTUs, as well as all OTUs above 1000 reads for a poplar and a pine ESB sample from 2016 (highlighted in red). The read abundance for the same OTUs in a beech and spruce sample from 2016, which were processed directly after the poplar and pine sample in the mill, are highlighted in red. The OTU abundances for beech and spruce samples from the same sites and processed in consecutive years are also shown. The total read abundance for each sample is shown below the tables (Reads).

• MDAR checklist

### Data availability

All raw reads are available in the Dryad Digital Repository (https://doi.org/10.5061/dryad.x0k6djhmp). The OTU table with metadata and qPCR results has been uploaded as supplementary files.

The following dataset was generated:

| Author(s) | Year | Dataset title | Dataset URL | Database and Identifier |
|---|---|---|---|---|
| Krehenwinkel H | 2022 | eDNA from archived leaves reveals no losses of α-diversity, but widespread community turnover and biotic homogenization as drivers of forest insect decline | https://doi.org/10.5061/dryad.x0k6djhmp | Dryad Digital Repository, 10.5061/dryad.x0k6djhmp |

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
