## [Editor Report]

This landmark study reveals novel temporal arthropod biodiversity insights that can be leveraged from environmental DNA traces, that have been cryopreserved on leaf tissue as part of a long-term monitoring scheme. The strength of the evidence underlying the major conclusions is convincing and limitations in the quantitative aspects of the data synthesis are acknowledged appropriately. The work will be of interest to a breadth of ecological practitioners.

---

## [Decision Letter]

**Decision letter after peer review:**

Thank you for submitting your article "Environmental DNA from archived leaves reveals widespread temporal turnover and biotic homogenization in forest arthropod communities" for consideration by *eLife*. Your article has been reviewed by 3 peer reviewers, and the evaluation has been overseen by a Reviewing Editor and Detlef Weigel as the Senior Editor. The following individuals involved in the review of your submission have agreed to reveal their identity: Rafael Valentin (Reviewer #1); Thomas Gilbert (Reviewer #3).

My colleagues have performed an excellent assessment of the breakthroughs and importance of this novel study and I will not paraphrase their valuable insights and feedback but will also join in the discussion.

It can be frequently observed that low diversity environmental DNA samples can be under sequenced using high throughput sequencing, whereas high diversity samples normally yield higher levels of sequence rates. By rarefying their analysis to 5000 reads (an approach that is increasingly not used, c.f. the McMurdy and Holmes debate https://journals.plos.org/ploscompbiol/article?id=10.1371/journal.pcbi.1003531), there is a risk that low diversity samples may be over-sampled and high diversity samples may be under-sampled. I appreciate that 5000 data points are likely to be appropriate when capturing leaf-based arthropod communities, but further evidence of the rarefaction, per sample coverage and rationale, would be valuable.

One of the conclusions of the research, according to qPCR of the 18S ribosomal DNA marker, is that (cellular) biomass appears to have decreased over time; a narrative that is coherent amongst the declining population trends that prevail in contemporary ecology. The challenge here is the use of the degenerate 18S marker, which is not linked to the interspecific copy number variation that will be prevalent amongst the different taxa in the study (e.g. taxon 1, 5 copies; taxon 2, 15 copies; taxon x, y copies…). A potentially equally plausible explanation for the declining copy number is that the copy number of the homogenised communities is lower than the older, non-homogenised communities. Without exhaustive per taxon calibration, the inability to compare interspecific abundance between taxa is one of the downsides of any metabarcoding study, that is focused on multicopy markers that differ either in copy number, or the amount of tandem repeats, as is the case with mitochondrial and nuclear ribosomal markers respectively. I suggest therefore that the investigators devise an additional test to measure, e.g., the level of association between qPCR copy number, time, and the level of homogenization (e.g. Jaccard dissimilarity) simultaneously in order to address this concern head-on.

In addition to my insights above, I will highlight the revisions that have been highlighted as essential amongst the review panel.

Essential revisions:

1) The biomass assay – Explore whether the decrease in qPCR (aka here as biomass) values is associated with homogenization, time, or other identifiable factors simultaneously (e.g. leaf characteristics) and discuss the results in an objective fashion.

2) The data have been rarefied to 5000 reads – is this enough and can the team clearly demonstrate their baseline data strategy? Would anything change by using either proportional data or using read depth as an offset on the total dataset (e.g. https://www.nature.com/articles/s42003-020-01562-4)?

3) Clarity on the nature of controls, decontamination procedures, and how these guided data filtering in the metabarcoding study.

4) Revisit the simplistic modelling and devise appropriate analyses that will take into consideration non-linear trends and different factors where appropriate.

5) Justify the replicability/robustness of the qPCR strategy.

6) Justify OTU picking strategy.

I would also invite the authors to address all the reviewer's comments via standard rebuttal letter and submission of track changed version of the original submitted text where possible. Where reviewers have different opinions, please discuss clearly your views in the context of your study and the evidence presented (cf. discussion of read numbers, qPCR, and abundance).

*Reviewer #1 (Recommendations for the authors):*

While reading through the methods I was at first deeply concerned that PCR errors remained within the zOTU table, but only later (in the statistical analysis section) did I see that OTUs with three or fewer reads were removed from the dataset. This should be moved up with the rest of the bioinformatic information to make it clear earlier and alleviate that concern.

I saw no mention of methods that would filter out reads, or samples, due to contamination levels. There is always some level of contamination that takes place in eDNA metabarcoding, and addressing this is paramount. Filtering out contaminants can range from removing contaminant reads in technical replicates to removing technical replicates entirely. This extends to the decision to use just two technical replicates for the metabarcoding portion of the study. While not bad, should technical replicates be filtered out it leaves just a single technical replicate to represent the sample, which isn't sufficient?

On line 185 only 413 OTUs were said to be used to assess temporal changes. Why were just these OTUs selected from the larger dataset? This should be explained and justified.

Were beads for the tissuelyzer and cryomil reused? If so, how were they decontaminated to ensure no contamination of subsequent samples took place? If they were not reused this should be made clear in the text.

What was the justification for a 3% OTU radius? Was this a precedent set from another paper, or was it a random selection? More detail here is needed.

*Reviewer #2 (Recommendations for the authors):*

I elaborate on my concerns regarding the sequence and statistical analyses below.

One of my greatest concerns is in the statistical analyses of the temporal trend. In most cases, the authors used linear models (LM) to test whether there is a temporal trend (i.e., increase or decrease). This is inappropriate because empirical time series often contain temporal autocorrelation structures and because the application of LM to such time series may often result in the false-positive detection of the trend. This means that LM can often detect a "significant" trend even in a random walk time series. This is a well-known issue, and the authors should apply a more appropriate statistical method, e.g., the autocorrelation model, state-space modeling, or some other methods, to judge whether there is a temporal trend.

Another concern is the use of read abundance as an explained variable in the statistical models. In L452-454, the authors claimed that the DNA copy numbers may be a proxy of biomass. I agree with this statement. However, sequence read abundance is not the DNA copy numbers and cannot be a proxy of abundance in most cases. If I understood correctly, the authors rarefied the sequence reads to 5000 reads for each sample, and the read abundance in the statistical model seems to be the relative abundance (e.g., if an OTU produced 500 reads in a sample, the relative abundance of the OTU in the sample is 10%). The authors quantified 18S DNA copy numbers of arthropods, so multiplying the relative abundance by the total 18S DNA copy numbers may produce a better proxy of the abundance of arthropods. I would recommend the authors reconsider the read abundance issue carefully.

*Reviewer #3 (Recommendations for the authors):*

Line 91. 312 ESB samples. Perhaps expand here on distribution through time etc? Could be part of Figure 1?

Line 125+ – stability of arthropod DNA comment. I'm actually surprised there is no higher diversity in the 8-year RT stored samples, as a result of post-mortem modifications to the DNA. Having said that does the PCR strategy negate that? If yes maybe worth stating this clearly. Actually in general given how critical the assumption that any zOTUs used represent true biological sequence variation (as opposed to PCR error, sequencing error, post mortem DNA damage) I advise the authors to early in the paper make it clear why they believe their zOTU estimates to be accurate.

Line 129. rDNA is typically used for many things. Ribosomal DNA. recombinant DNA. Even relative DNA. Maybe for clarity here state 'information on relative arthropod 18s rDNA copy' then there is no uncertainty.

Figure 5F's label is confusing (e.g. Does < 2020 mean 2010-2020 or all years before 2020 etc?). I suspect the former but that's not how it reads as written in the figure.

Can the authors elaborate on when the leaves were collected in the year? In line 287 the authors state it was the same time point every year. How is that shaped by the time of year the leaves come out every year? (Which must fluctuate a lot with annual weather variation). Also in line 257, the authors state the leaves represent a 'fairly broad phenological window'. Can this be elaborated on?

Line 287/288. It seems odd to write 'a defined amount' of both leaf and number of trees, and then give values prefixed by '>'. And then say 'defined number of branches from each tree' but then not give a value. I think I know what the intended meaning is, but perhaps consider rewording this sentence! Also, I assume the exact numbers are somewhere – maybe refer to where at this point?

Methods in general, it took me a long time to work out exactly how the samples were collected and processed, and I'm still not sure. If I understand it correct – leaves were originally picked from trees immediately into liquid nitrogen, and then shortly after ground to a powder? Thus all happening years/decades ago? If correct please consider re-reading the text assuming you know nothing about the history and add any small clarifications that might be needed so the reader can immediately jump to this conclusion. If however, I misunderstand…then again the text needs clarification.

Line 295 I assume the cryomill was somehow decontaminated between sample batches? Please clarify how.

Ca Line 365. Please state clearly if extraction replicates were made on the samples used in the full experiment, (or not). It's not clear to me as written now. I can see they were used in the sample mass trial and perhaps given the replicate dissimilarity results were ok for the PCR and extraction replicate the decision was taken to not do replicates on the large scale? If so please state it clearly.

---

## [Author Response]

My colleagues have performed an excellent assessment of the breakthroughs and importance of this novel study and I will not paraphrase their valuable insights and feedback but will also join in the discussion.It can be frequently observed that low diversity environmental DNA samples can be under sequenced using high throughput sequencing, whereas high diversity samples normally yield higher levels of sequence rates. By rarefying their analysis to 5000 reads (an approach that is increasingly not used, c.f. the McMurdy and Holmes debate https://journals.plos.org/ploscompbiol/article?id=10.1371/journal.pcbi.1003531), there is a risk that low diversity samples may be over-sampled and high diversity samples may be under-sampled. I appreciate that 5000 data points are likely to be appropriate when capturing leaf-based arthropod communities, but further evidence of the rarefaction, per sample coverage and rationale, would be valuable.

We have now expanded on this in the Methods, added more data and performed some additional analyses on the read coverage question. 5000 reads from two replicates each per sample should indeed achieve saturation of diversity in this case (Figure 2—figure supplement 4). We went for an equal coverage approach for all samples, which means the coverage for each sample was based on the lowest covered sample. Hence, most samples have considerably more reads than 5000. To address potential biases of this approach, we repeated the analysis with unrarefied data, which led to 21,676 reads per replicate sample on average. The results essentially remain unchanged (see Methods: Statistical Analysis; Suppl. File 2), with diversity of the two datasets highly correlated (R^2^=0.95). This analysis can be repeated by all readers based on the raw read data, which allows for much deeper coverage. We have included this in the Methods now (lines 581-588).

In addition, we explored the effect of including triplicate samples and an even deeper coverage in two complete timeseries of 42 beech samples. Here, we sequenced triplicates of every sample, resulting in an average coverage of 78,667 reads per sample. Again, the result remained unchanged compared to the reduced coverage dataset (lines 435-439).

In addition, we generated a second dataset for most samples with another primer pair, which also confirmed our findings from the first dataset. The fact that the second dataset recovered essentially identical patterns of α and β diversity also suggest the validity of our dataset (Figure 2—figure supplement 5).

We liked the recommendation by Reviewer 1 to combine data from qPCR and metabarcoding for a copy number-based rarefaction. We have also included this analysis now and find no change to our results (lines 583-588).

Considering this background, we are confident that the result we aim to achieve, namely testing for a temporal change of richness or arthropod community turnover, is not affected by our read sampling strategy.

One of the conclusions of the research, according to qPCR of the 18S ribosomal DNA marker, is that (cellular) biomass appears to have decreased over time; a narrative that is coherent amongst the declining population trends that prevail in contemporary ecology. The challenge here is the use of the degenerate 18S marker, which is not linked to the interspecific copy number variation that will be prevalent amongst the different taxa in the study (e.g. taxon 1, 5 copies; taxon 2, 15 copies; taxon x, y copies…). A potentially equally plausible explanation for the declining copy number is that the copy number of the homogenised communities is lower than the older, non-homogenised communities. Without exhaustive per taxon calibration, the inability to compare interspecific abundance between taxa is one of the downsides of any metabarcoding study, that is focused on multicopy markers that differ either in copy number, or the amount of tandem repeats, as is the case with mitochondrial and nuclear ribosomal markers respectively. I suggest therefore that the investigators devise an additional test to measure, e.g., the level of association between qPCR copy number, time, and the level of homogenization (e.g. Jaccard dissimilarity) simultaneously in order to address this concern head-on.

We agree that we cannot fully rule out the possibility that a change in copy number is due to shifts in community composition, and we have already addressed this possibility in the discussion (lines 257-260).

The 18S we used here is indeed a multi-copy marker, and shifts in community composition could also lead to shifts in copy number. This could mimic the pattern of copy number loss over time. However, we have tested our assay quite extensively. The test included diverse mock communities in a very pronounced dilution series (1:10,000), which still suggested that it predicts copy number changes accurately and with a high efficiency (Figure 3). The 1:10,000 dilution we used to estimate qPCR efficiency is also far lower than arthropod copy number in any actual ESB sample, yet with this assay we accurately predicted the amount of arthropod DNA. We also have found 18S to reflect coarse changes to abundance quite well in two previous studies, which we cited in the manuscript (Krehenwinkel et al. 2017 and 2019).

A strong change in copy number due to change in taxonomic composition would likely be found if turnover affects divergent taxa with very different copy number. This is contrary to our findings. We find that usually closely related taxa replace each other. E.g., a mining moth gets replaced by another one (see Figure 5—figure supplement 1).

The test for an association of homogenization and/or turnover with copy number is a very good idea, and we have now implemented this in our analysis. We have explored the correlation between community dissimilarity and the difference in copy number between sampling years (Figure 5—figure supplement 2). With the exception of one urban site, we do not find a clear association. In particular the beech forest sites, which show increasing homogenization over time, do not show an association of copy number differences and turnover. We thus believe that the decrease in copy number is better explained by a loss of biomass than simple turnover. This is also in line with recent work on insect decline, suggesting that abundance losses of common species can drive insect decline. However, we still cannot rule it out entirely and hence left the discussion of the topic (lines 257-260).

On a side note: Just last week I saw a very interesting presentation at a conference supporting our findings. The authors (Samu et al.) found stable richness of forest spider communities over time, but a loss of biomass as a general pattern for the studied spider communities.

In addition to my insights above, I will highlight the revisions that have been highlighted as essential amongst the review panel.Essential revisions:1) The biomass assay – Explore whether the decrease in qPCR (aka here as biomass) values is associated with homogenization, time, or other identifiable factors simultaneously (e.g. leaf characteristics) and discuss the results in an objective fashion.

We have now addressed this in the Methods (lines 626-629), Results (lines 197-198) and Discussion (lines 259-260). We did not find an association between copy number and β diversity. Assuming that turnover is driving the changes in copy number, we would find an association of β diversity and dissimilarity in copy number. With the exception of one site, this was not found (Figure 5—figure supplement 2).

Also, the observed turnover mostly affects closely related taxa. No community tipping point or pronounced change to the higher-level taxonomic compositions of the community is found in our data. Pronounced changes in copy number would rather be expected with replacement of distantly related taxa. This also makes copy number changes due to turnover less likely. Our mock community assay also shows that the qPCR assay quite accurately recovers copy number changes even in quite divergent communities.

We can also rule out changes in leaf characteristics as drivers of changes to relative copy number. The specimen bank collects biometric data for most its samples, including leaf weight, which has not changed over time in most sampling sites (lines 310-311).

Considering this background, we consider an actual loss of biomass a more likely explanation. Though we cannot fully rule out the effect of turnover, an overall loss of biomass affecting many common species would provide a simple explanation for the stable richness but drop of biomass.

2) The data have been rarefied to 5000 reads – is this enough and can the team clearly demonstrate their baseline data strategy? Would anything change by using either proportional data or using read depth as an offset on the total dataset (e.g. https://www.nature.com/articles/s42003-020-01562-4)?

We have now added several additional analyses and new data to underline the reliability of our data, which we address in the “Statistical analysis” part of the Methods (lines 581-588). 5,000 reads per replicate appear to suffice to saturate recovered diversity, as seen in this rarefaction curve (Figure 2—figure supplement 4). To make sure we did not undersample the communities, we repeated the analysis with unrarefied data, which led to 21,676 reads per replicate sample on average. The results essentially remain unchanged, with diversity highly correlated (R^2^=0.95). This analysis can be repeated by all readers based on the raw read data, which allows for deeper coverage than 5,000.

In addition, we explored the effect of including triplicate samples (which we newly generated) and an even deeper coverage in two complete timeseries of 42 beech samples. Here we sequenced triplicates of every sample, resulting in an average coverage of 78,667 reads per sample. Again, the result remained unchanged compared to the reduced coverage dataset (lines 435-439).

In addition, we generated a second dataset for most samples using a second primer pair, which also confirmed our findings from the first dataset. Using a completely different primer set most likely significantly increased the recovered taxonomic diversity. The fact that the second dataset recovered essentially identical patterns of α and β diversity also suggests the validity of our dataset (Figure 2—figure supplement 5).

We liked the idea of proportional sampling and have also incorporated a novel analysis on this (lines 583-588). We have used the relative copy number as a proxy to rarefy our dataset; i.e., the read coverage was based on relative arthropod copy number in a sample. Even incorporating the overall declining copy number in our rarefaction did not result in a temporal loss of richness for the studied communities, additionally underling the robustness of our data.

3) Clarity on the nature of controls, decontamination procedures, and how these guided data filtering in the metabarcoding study.

We have expanded on this in the Methods section of the manuscript, particularly under “Tree samples of the German Environmental Specimen Bank – Standardized time series samples stored at ultra-low temperatures”, “Test for DNA carryover in the cryomill” and “Statistical analysis”. We added details on extraction and PCR controls and more information on decontamination protocols assuring the sterility of the sampling procedure of the ESB (lines 303-304, 316). The ESB sampling is only performed with sterile equipment, which is thoroughly cleaned between sampling, making cross-contamination unlikely. The thorough cleaning and decontamination are also done for the cryomill.

We have now also included a new test to explore the possibility of contamination by the cryomill: “Test for DNA carryover in the cryomill” in the Methods section (lines 448-464). We have collected information on the milling schedule of samples processed between 2016 and 2019. We then identified tree samples that were processed consecutively in the cryomill. Assuming carryover is an issue, tree samples that were processed consecutively should show a significantly reduced β diversity between each other than between samples from the same site processed in different years. This effect was not found (Figure 2—figure supplement 3). We also tested single species carryover in more detail. This was done with a poplar and beech sample as well as a spruce and pine sample processed on two consecutive days. Assuming there is carryover in the mill, tree species-specific arthropod species should show signs of cross-contamination between the samples. This should only affect consecutively processed samples, but not samples from the same site processed in different years. The according samples were sequenced in triplicates to deep coverage (78,667 reads per sample on average) and the read abundances of the most common tree-species-specific arthropods measured. No sign of cross-contamination was found. This can be seen in the new (Suppl. File 3).

4) Revisit the simplistic modelling and devise appropriate analyses that will take into consideration non-linear trends and different factors where appropriate.

We have reworked statistical analysis now accounting for autocorrelation as reported under “Statistical analysis” in the Methods (lines 629-644). As a reviewer rightfully pointed out that abundance changes should not follow linear trends only, we have now used a more appropriate non-parametric test to explore species with significant increase or decline for the single species abundance change analysis. We have also used the qPCR-corrected read count for this analysis (Figure 5 A and B). The new analyses resulted in no overall change of the results.

5) Justify the replicability/robustness of the qPCR strategy.

We have now explained this in more detail in the methods, results and discussion, particularly under “Detection of relative arthropod DNA copy number using quantitative PCR” (lines 504-564). We based our idea of using nested qPCR on a paper (Tran et al. 2014), which we cite in the manuscript. This works shows that nested qPCR has a considerably higher sensitivity than standard qPCR, while retaining a very high accuracy. This is also what we found with extensive testing in our assay.

We tested the efficiency of the qPCR assay in a 1:10000 dilution series of two ESB tree DNA extracts. Both showed a very high efficiency from 1X to 1:10000 dilution. Changes in copy number of plants and insects are thus very reliably recovered by our assay. The CT values at a 1:10000 dilution are considerably higher (CT_1:10000_~30) than what we found in our experiments with the actual ESB extracts (average CT plant assay = 15.7; average CT insect assay = 21.3). Hence our assay is accurate and efficient at the whole range of arthropod DNA abundance found in our samples.

The accuracy of the assay is also shown in our mock community experiment, where spiked-in community DNA of seven highly distinct mock communities showed very comparable changes in copy number (Figure 3).

The technical replicates of our qPCR are very comparable with each other, showing a CT SD of ~0.10 on average. Also, all qPCRs were run along with a negative control to make sure contamination did not affect the outcome. Technical replicate CT values are now provided in the Suppl. File 2.

6) Justify OTU picking strategy.

We have explained our strategy in more detail in the Methods under “Sequence processing” (lines 488-494). We used three different approaches for OTU picking. The first two are implemented in USEARCH. Using cluster_otus, we generated 3 % OTUs; using unoise3, we generated zero radius OTUs corresponding to haplotypes. The first should approximate species richness, while the latter will allow us to explore genetic variation within species. To make the assessment of genetic variation more stringent, we have also included additional quality filtering criteria to call zOTUs, which we report on in the methods.

The German Barcode of Life for arthropods is quite complete, hence, we could ground-truth the species assignment by 3 % OTUs quite well. We found the 3% OTUs to over-split many species, e.g., a single species consists of various 3 % OTUs. We hence included ptp clustering (Zhang et al. 2013) as third OTU picking strategy. PTP clusters approached actual species assignments by BOLD much better than 3 % radius OTUs; hence, we used only zOTUs and PTP OTUs for further analysis. However, patterns of diversity were highly correlated for all three OTU picking strategies as we report in the Methods and in Suppl. File 2.

I would also invite the authors to address all the reviewer's comments via standard rebuttal letter and submission of track changed version of the original submitted text where possible. Where reviewers have different opinions, please discuss clearly your views in the context of your study and the evidence presented (cf. discussion of read numbers, qPCR, and abundance).Reviewer #1 (Recommendations for the authors):While reading through the methods I was at first deeply concerned that PCR errors remained within the zOTU table, but only later (in the statistical analysis section) did I see that OTUs with three or fewer reads were removed from the dataset. This should be moved up with the rest of the bioinformatic information to make it clear earlier and alleviate that concern.

This information has now been added to the “Sequence processing” section (lines 573-575).

I saw no mention of methods that would filter out reads, or samples, due to contamination levels. There is always some level of contamination that takes place in eDNA metabarcoding, and addressing this is paramount. Filtering out contaminants can range from removing contaminant reads in technical replicates to removing technical replicates entirely. This extends to the decision to use just two technical replicates for the metabarcoding portion of the study. While not bad, should technical replicates be filtered out it leaves just a single technical replicate to represent the sample, which isn't sufficient?

As mentioned above, we have expanded the section on data cleaning. We used both technical replicates after careful evaluation of contamination with the help of controls. We made sure both technical replicates showed predictable and comparable taxon composition and were not affected by contamination. We have now also included a dataset of triplicates (lines 435-439) and an unrarefied dataset (lines 581-588) to show that additional replication/deeper sequencing does not change the recovered biodiversity patterns (as reported under “Statistical analysis”). The fact that we used a second primer pair for our analysis, which also supports the same biodiversity patterns as our first primer pair, additionally gives us high confidence in the reliability of our data (Figure 2—figure supplement 5).

On line 185 only 413 OTUs were said to be used to assess temporal changes. Why were just these OTUs selected from the larger dataset? This should be explained and justified.

We address this issue in Methods: “Statistical analysis” (lines 594-598). As mentioned above, we excluded OTUs to make sure to only include reproducibly occurring taxa from the timeseries. Using rare OTUs, which for example only occur with a few reads or only a single time in our dataset, makes it more likely that we base our analysis on artefacts. A taxon occurring a single time in the time series may show a pattern of decline if it occurs only in the first datapoint of the timeseries. Also, a taxon represented by very few reads is more likely to originate from contamination. This is why we chose a minimum of 3 occurrences and 100 reads to score a taxon. We only included time series of 10 years or longer as changes in shorter time series may also be part of natural abundance fluctuations instead of an actual decline.

Were beads for the tissuelyzer and cryomil reused? If so, how were they decontaminated to ensure no contamination of subsequent samples took place? If they were not reused this should be made clear in the text.

The tissuelyzer beads were new and not reused, which we report now in the Methods (lines 394-396). The cryomill is a very large device, which uses titanium cylinders to grind samples. These cylinders are reused after thorough cleaning and decontamination, which is performed on all parts of the mill after each run. As mentioned above, we have now also included an analysis that shows there is no signal of between-sample carryover between milling events. We have added details to the text to clarify this (“Test for DNA carryover in the cryomill”, lines 448-464; Figure 2—figure supplement 3, Suppl. File 3).

What was the justification for a 3% OTU radius? Was this a precedent set from another paper, or was it a random selection? More detail here is needed.

We have now expanded on our OTU picking strategy in the text and explained why different OTUs were used (see ”Sequence processing”, lines 488-494). We used three different OTU picking strategies:

1. Zero radius OTUs or zOTUs, which represent haplotypic variation within species. We have now added a stringent filter to filter these zOTUs to calculate within-OTU diversity as a measure of genetic diversity.

2. 3 % radius OTUs are very commonly used in metabarcoding studies and usually assumed to represent species-level diversity. We initially also used 3 % radius OTUs for this purpose. The barcode reference library for German arthropods is very complete for many groups, hence we can reliably assign species status to many OTUs. But when comparing the 3% OTUs to the barcode reference library, we found that 3 % OTUs very commonly over-split species. E.g., a single species consisted of several OTUs. We have hence used a third clustering approach:

3. Ptp clustering commonly merged several OTUs into a single cluster. These clusters hence much better represented actual species diversity than 3 % OTUs. We therefore used ptp-based OTUs to represent species diversity.

However, all three OTU clustering approaches resulted in highly correlated diversity metrics (Suppl. File 2). So, for the general message on diversity change in German ecosystems, the OTU clustering approach does not seem to have a detectable effect.

Reviewer #2 (Recommendations for the authors):I elaborate on my concerns regarding the sequence and statistical analyses below.One of my greatest concerns is in the statistical analyses of the temporal trend. In most cases, the authors used linear models (LM) to test whether there is a temporal trend (i.e., increase or decrease). This is inappropriate because empirical time series often contain temporal autocorrelation structures and because the application of LM to such time series may often result in the false-positive detection of the trend. This means that LM can often detect a "significant" trend even in a random walk time series. This is a well-known issue, and the authors should apply a more appropriate statistical method, e.g., the autocorrelation model, state-space modeling, or some other methods, to judge whether there is a temporal trend.

As mentioned above, we have reanalyzed our data accordingly and report this in the Methods and Results. The reported patterns essentially remain unchanged after the new analysis.

Another concern is the use of read abundance as an explained variable in the statistical models. In L452-454, the authors claimed that the DNA copy numbers may be a proxy of biomass. I agree with this statement. However, sequence read abundance is not the DNA copy numbers and cannot be a proxy of abundance in most cases. If I understood correctly, the authors rarefied the sequence reads to 5000 reads for each sample, and the read abundance in the statistical model seems to be the relative abundance (e.g., if an OTU produced 500 reads in a sample, the relative abundance of the OTU in the sample is 10%). The authors quantified 18S DNA copy numbers of arthropods, so multiplying the relative abundance by the total 18S DNA copy numbers may produce a better proxy of the abundance of arthropods. I would recommend the authors reconsider the read abundance issue carefully.

As mentioned above, we did not use read abundance, but qPCR-based copy number as a measure for abundance. We developed a very robust qPCR assay for this purpose. As suggested by the reviewer, we have rarefied our OTU tables using qPCR-based copy number instead of equal coverage (see “Statistical analysis”). Using this new dataset did not change the recovered diversity patterns. We have also repeated this exercise for the single species abundances. Here we also find no general change of the observed patterns (Figure 5).

Reviewer #3 (Recommendations for the authors):Line 91. 312 ESB samples. Perhaps expand here on distribution through time etc? Could be part of Figure 1?

We feel this may overload the figure a bit.

Line 125+ – stability of arthropod DNA comment. I'm actually surprised there is no higher diversity in the 8-year RT stored samples, as a result of post-mortem modifications to the DNA. Having said that does the PCR strategy negate that? If yes maybe worth stating this clearly. Actually in general given how critical the assumption that any zOTUs used represent true biological sequence variation (as opposed to PCR error, sequencing error, post mortem DNA damage) I advise the authors to early in the paper make it clear why they believe their zOTU estimates to be accurate.

Thank you for this interesting comment; we had not thought about the possibility of chemical degradation here. We are aware of studies on fossil DNA, were this is certainly a big issue. Do you think this could affect samples only a few decades old? The dried homogenates we are using so far were only ~ 10 years old, so we would be surprised if they showed chemical degradation. But we are currently exploring even older dried homogenate samples and indeed find a higher diversity in the oldest samples (> 20 years) than the younger ones. We will now carefully explore whether chemical degradation has an effect here. We have added a sentence and citation in the Discussion urging for care in analysis of dried samples (lines 241-243). Luckily, we are basing our work here on samples that have been stored on liquid nitrogen, hence there is no possibility for chemical modification.

When it comes to richness and β diversity, zOTUs and OTUs show highly replicated patterns. There does not seem to be an inflation of diversity here for zOTUs. However, we have now added more details on the picking of zOTUs for the analysis of genetic diversity in the methods (“Statistical analysis”, lines 611-615). We have included several selection steps to use them, as elaborated above.

Line 129. rDNA is typically used for many things. Ribosomal DNA. recombinant DNA. Even relative DNA. Maybe for clarity here state 'information on relative arthropod 18s rDNA copy' then there is no uncertainty.

Has been changed (line 129).

Figure 5F's label is confusing (e.g. Does < 2020 mean 2010-2020 or all years before 2020 etc?). I suspect the former but that's not how it reads as written in the figure.

We have now clarified this in the figure caption (lines 227-228).

Can the authors elaborate on when the leaves were collected in the year? In line 287 the authors state it was the same time point every year. How is that shaped by the time of year the leaves come out every year? (Which must fluctuate a lot with annual weather variation). Also in line 257, the authors state the leaves represent a 'fairly broad phenological window'. Can this be elaborated on?

We have added details in the Methods (lines 301-303). The aim of the ESB is to span the complete growing period of the leaf. That means beech leaves, for example, are collected between late August and early September. We have included the month of sampling in our statistical analysis. Over the whole time series, there is no trend of shifts in sampling events. The events are randomized on a time window between mid-August and mid-September. No effect of sampling month on diversity patterns was found.

The broad phenological window is based on the fact that many arthropods that live on leaves will only be outside of the leaf to mate for a few weeks in the year. During this time, you could catch them in a Malaise trap. The rest of the year, they spend in or on the leaves of their host plants. A good example is the spruce gall midge, which only flies for a few weeks in May, but can be detected in the leaves throughout the year. We have explained this a little more in the text (lines 234-235).

Line 287/288. It seems odd to write 'a defined amount' of both leaf and number of trees, and then give values prefixed by '>'. And then say 'defined number of branches from each tree' but then not give a value. I think I know what the intended meaning is, but perhaps consider rewording this sentence! Also, I assume the exact numbers are somewhere – maybe refer to where at this point?

We have expanded on this and provided more details in the Methods (“Tree samples of the German Environmental Specimen Bank – Standardized time series samples stored at ultra-low temperatures”, lines 301-308). The protocol is always exactly the same for each site. All variation of the protocol has been incorporated into the statistical model.

Methods in general, it took me a long time to work out exactly how the samples were collected and processed, and I'm still not sure. If I understand it correct – leaves were originally picked from trees immediately into liquid nitrogen, and then shortly after ground to a powder? Thus all happening years/decades ago? If correct please consider re-reading the text assuming you know nothing about the history and add any small clarifications that might be needed so the reader can immediately jump to this conclusion. If however, I misunderstand…then again the text needs clarification.

We have explained the sampling in more detail in the Methods (“Tree samples of the German Environmental Specimen Bank – Standardized time series samples stored at ultra-low temperatures”, lines 314-319). The ESB is unique in the whole world in that samples really go into the nitrogen in the field and never leave the nitrogen again until we analyze them.

Line 295 I assume the cryomill was somehow decontaminated between sample batches? Please clarify how.

We have added more information on decontamination protocols implemented by the ESB in the Methods (“Tree samples of the German Environmental Specimen Bank – Standardized time series samples stored at ultra-low temperatures”, line 316; and especially “Test for DNA carryover in the cryomill”, lines 448-464). The ESB sampling is only performed with sterile equipment, which is thoroughly cleaned between sampling, making cross-contamination unlikely. The thorough cleaning and decontamination are also done for the cryomill.

However, we have now included a new test to explore the possibility of contamination by the cryomill (Figure 2—figure supplement 3 and Suppl. File 3). We have collected information on the milling schedule of samples processed between 2016 and 2019. We have then identified tree samples that were processed consecutively in the cryomill. Assuming carryover is an issue, tree samples that were processed consecutively should show a significantly reduced β diversity between each other than between samples from the same site processed in different years. This effect was not found. We also tested single species carryover in more detail. This was done with a poplar and beech sample as well as a spruce and pine sample processed on two consecutive days. Assuming there is carryover in the mill, tree species-specific arthropod species should show signs of cross-contamination between the samples. This should only affect consecutively processed samples, but not samples from the same site processed at different years. The according samples were sequenced in triplicate to deep coverage (~80,000 reads per sample on average) and the read abundances of the most common tree-species-specific arthropods measured. No sign of cross-contamination was found, as seen in the new Figure 2—figure supplement 3 and Suppl. File 3.

Ca Line 365. Please state clearly if extraction replicates were made on the samples used in the full experiment, (or not). It's not clear to me as written now. I can see they were used in the sample mass trial and perhaps given the replicate dissimilarity results were ok for the PCR and extraction replicate the decision was taken to not do replicates on the large scale? If so please state it clearly.

We now state this explicitly in the Methods (lines 432-435). Considering the similar recovery of taxa from PCR and extraction replicates, we decided to run PCR replicates as technical replicates and no extraction replicates. We recently performed several replication experiments with different ESB homogenates, underlining the high efficiency of the cryomill in homogenizing the samples. For the well homogenized samples, only a small subset of homogenate is needed to saturate diversity. We find this across different ESB samples and taxa. PCR and/or extraction replication will essentially have a similar effect: more rare taxa will be picked up with replication. However, even after 2 replicates, we find the samples to show acceptable saturation, as we show that additional replication does not change our results.